# First Identification and Quantification of Detached Tip Vortices Behind a WEC Using Fixed Wing UAS

Moritz Mauz[1], Alexander Rautenberg[1], Andreas Platis[1], Marion Cormier[2], and Jens Bange[1]

[1]Centre for Applied Geoscience, Eberhard Karls University of Tübingen, 72074 Tübingen, Germany
[2]Institute of Aerodynamics and Gas Dynamics, University of Stuttgart, 70569 Stuttgart, Germany

**Correspondence:** Moritz Mauz (moritz.mauz(at)uni-tuebingen.de)

**Abstract.** In the present study, blade-tip vortices have been experimentally identified in the wake of a commercial wind turbine using the MASC Mk 3 (Multi-purpose Airborne Sensor Carrier Mark 3) UAS(Unmanned Aircraft System) of the University of Tuebingen. By evaluation of the wind components, detached blade-tip vortices were identified in the time series. From these measurements the circulation and core radius of a pair of detached blade-tip vortices is calculated using the Burnham-Hallock (BH) wake vortex model. The presented data were captured under a dominating marine stratification about $2\,\mathrm{km}$ from the North Sea coast line with northern wind direction. The measured vortices are also compared to the analytical solution of the BH model for two vortices spinning in opposite direction. The model has its origin in aviation, where it describes two aircraft wake vortices spinning in opposite direction.

An evaluation method is presented to measure detached tip vortices with a fixed wing UAS. The BH model will be used to describe wake vortex properties behind a wind energy converter (WEC). The circulation and core radius of detached blade-tip vortices will be calculated. Also a proposition of the model for WEC wake evaluations will be made to describe two independent co-rotating vortices. Quantifying blade-tip vortices helps to understand the process of vortices detaching from a rotor blade of a wind turbine, their development in the wake until finally dissipating in the far wake and contributing to overall atmospheric turbulence. This is especially interesting for set-ups of numerical simulations when setting the spatial resolution of the simulation grid.

## 1 Introduction

The wind energy sector has been growing world wide for decades and the produced power from wind energy is still growing. Not only the amount of installed wind energy converters (WECs) is increasing but also the capacity of a single turbine. Also the field of application has widely increased with WEC. There are systems available for homogeneous terrain, off or near shore, or even complex terrain with a high amount of additional turbulence stress that is induced onto the wind turbine's blades.

A modern off-shore WEC delivers up to 12 MW of power in ideal conditions. In wind energy research numerical simulations of the wind velocity field of a WEC and its produced turbulence are important tools that give valuable informations. Pressure and velocity distributions around a turbine blade and nacelle as well as in the wake can be studied. A numerical model increases its validity when it is backed by real world in-situ data. Once measured data has revealed some possible tweaks and

5 enhancements to a model, improvements can be made and flow back into the (e.g. numerical) model. Numerical simulation might underestimate peak vorticity and radii of wake vortices, especially when the grid size of the simulation is not sufficient (Kim et al., 2016). Another way of studying WEC wakes are wind tunnel experiments that try to recreate wake patterns in a smaller scale (e.g. Bartl et al. (2012) or Vermeer (1992)). While in the early days of wind tunnel experiments the wake has been visualised by smoke trails, PIV (Particle Image Velocimetry) measurements have increased the resolution and accuracy of

10 wind tunnel experiments drastically (e.g. resolving Reynolds shear stress and turbulent kinetic energy, Zhang et al. (2012)). But a common issue with wind tunnel measurements is that they usually suffer from scaling problems (Wang et al., 1996). Remote sensing techniques like LIDAR have also found their way into WEC wake evaluations. Various measurement strategies were developed to visualise WEC wakes e.g. in complex terrain (Barthelmie et al., 2018). Typical LIDAR scans provide a long term measurement of a probed volume or plane. The spatial resolution ($25 - 50$ m), however, is comparably coarse. LIDARs can

provide a continuous monitoring of WEC wake structures (e.g. wake centre, direction and wind velocity deficit) (Bodini et al., 2017) in homogeneous or even in complex terrain (Wildmann et al., 2018). Short-range continuous-wave LIDARs provide even higher spatial resolution for short focal distances and have been applied in WEC wake measurements (Menke et al., 2018), yet these measurements can still not resolve blade tip vortices. UAS (Unmanned Aircraft System) measurements can provide in-situ line measurements, covering a small volume but with a high temporal and spatial resolution in (deca-) centimetre range.

The coverage of these scales is important to measure detached tip vortices in the near wake of a WEC.

A WEC, especially in a stable marine ABL (atmospheric boundary layer), acts as a turbulence generator. The added turbulence has two main sources. On the one hand the increased wind shear in the wake that results from the wind deficit in the near wake and the low pressure bulb that develops behind the WEC nacelle. On the other hand turbulence is created by expansion and dissipation of detached blade-tip vortices that transfer their kinetic energy to the surrounding flow. A proper understanding

of these vortices and their induced load onto the converter blade is of great importance for future enhancement of life span and working loads of wind energy converters in wind farms. Blade-tip vortices follow a helical pattern into the wake, detaching from each converter blade. These detached eddies can be measured with the mounted five-hole-probe on the MASC UAS. Subramanian et al. (2015) detected tip vortices via pressure fluctuations qualitatively in a flight pattern along the wake, also using a small UAS. In this study an evaluation method is presented to measure the core radius $r_\mathrm{c}$, circulation $\Gamma$ and maximum

tangential velocity $V_\mathrm{t,max}$ of a tip vortex using in-situ wind measurements from UAS flights perpendicular to the mean wind velocity.

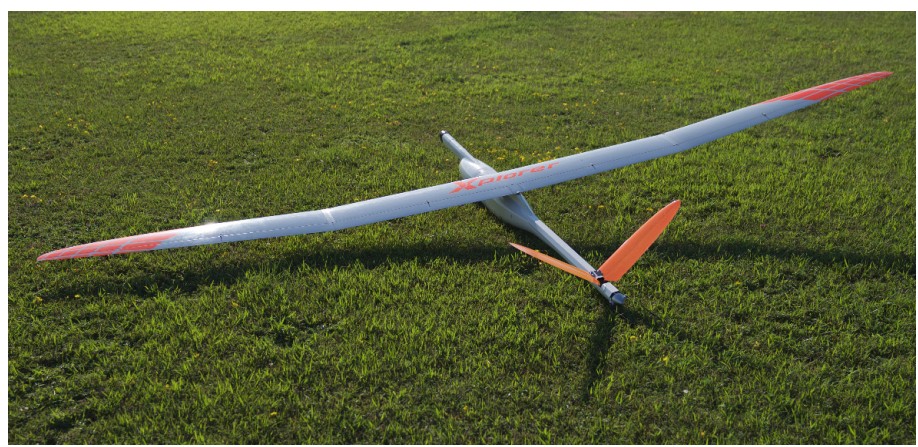

**Figure 1.** Research UAS MASC Mk 3 shortly before lift-off. (Photo taken by the author).

**Table 1.** Characteristics of the MASC Mk 3 UAS at the HeliOW campaign.

| | |
|---|---:|
| wingspan | 4 m |
| total weight | $\approx 7$ kg |
| sci. payload | $\approx 1$ kg |
| cruising speed | $19 \text{ m s}^{-1}$ |
| endurance | up to 2.5 h |
| propulsion | electrical pusher engine |
| take-off | bungee or winch |

## 2  Measurement system and measurement site

### 2.1  Research aircraft

The research UAS MASC Mk 3 (cf. Fig. 1 and Tab. 1) is a fixed wing airborne measurement system of the University of Tübingen that has been used in several measurement campaigns and has been described by Wildmann et al. (2014a, b). The third iteration of this platform features some changes to the fuselage. The electrical pusher motor has been moved from a centre position behind the wings to the tail, accelerating the aircraft along the centre axis and increasing flight stability. The MASC Mk 3 system allows in-situ high frequency measurements of the atmospheric flow and its transported properties. A detailed description of the improved UAS and its instruments can be found in Rautenberg et al. (2019b). The latest iteration MASC Mk 3 is using an improved IMU (Inertial Measurement Unit) and positioning system.

Aside the changes in fuselage design, the former ROCS autopilot operating on the MASC Mk 2 system has been changed to the Pixhawk 2.1 autopilot. This is an independent open-hardware and open-source autopilot project (Pixhawk-Organisation).

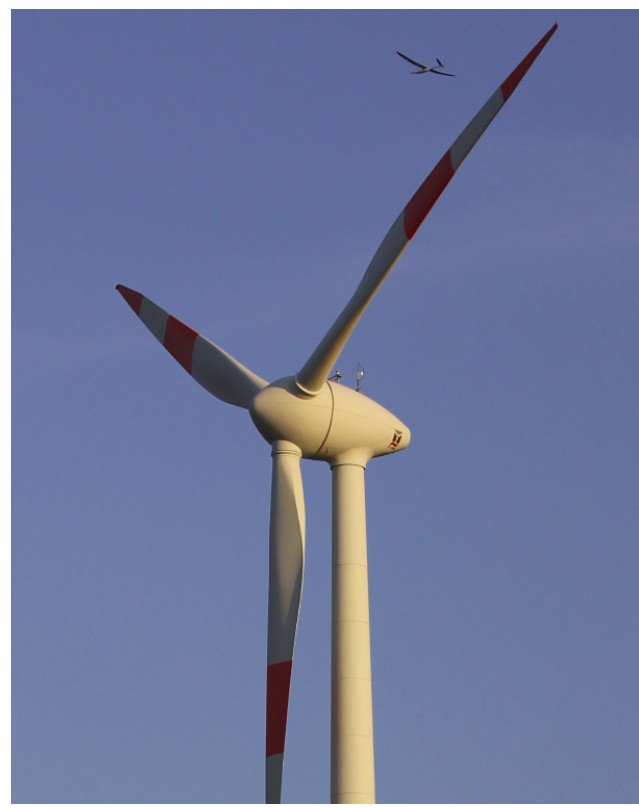

**Figure 2.** Research UAS MASC Mk 3 in front of an Enercon wind energy converter at the Jade Wind Park in July 2018. (Photo taken by the author).

## 2.2 Measurement site

Figure 3 shows the location of the measurement site in the north-west of Germany and the flight tracks of the MASC Mk 3 UAV around the Enercon E-112 converter. Both tracks are part of a rectangular flight pattern around the WEC in anti-clockwise direction. For the wake data evaluation only the data captured along the southern flight tracks (orange path in Fig. 3) are used.

5   The E-112 WEC is the most powerful converter in the Jade-Wind-Park north of Wilhemshaven, Germany. The particular converter is a former near-shore prototype with a rotor diameter D of 114 m delivering up to 4.5 MW of electrical power and thus comparable to an actual off-shore WEC. The Jade-Wind-Park is located about 2 km from the North Sea coast line and a maritime influence in the wind profile can be expected.

Apart from surrounding WECs (to the south of the E-112 WEC) power lines to the east and north and industrial buildings to

10   the north and north-east (not in the picture) restricted the flight path to the ones depicted in Fig. 3.

    For this study of the near wake of a WEC, the wind turbine described above, has been chosen. This specific converter and its location near the coast is comparable with off-shore converters in marine flow which was a requirement when choosing the WEC. The measurements are part of the HeliOW project, in which the atmospheric turbulence in front of and in the wake

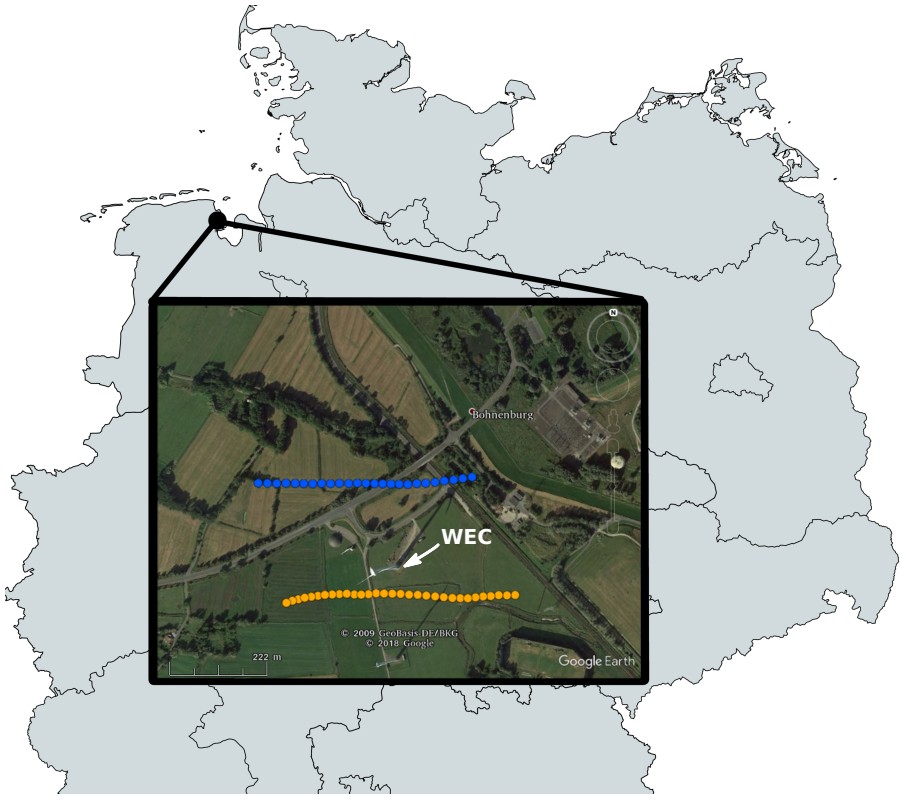

**Figure 3.** Location of the E-112 WEC in the north-west of Germany north of Wilhelmshaven near the North Sea coast. MASC flight tracks in front (blue) and in the wake (orange) of the E-112 with northern main wind direction (5 degree north that day). On the Google Earth image the WEC is oriented toward south-easterly wind direction. Map created with mapchart.net

of a WEC are the foundation of a chain of numerical simulations. The goal of the project is also to determine safe helicopter flight paths in off-shore wind energy parks. The numerical simulation chain also includes CFD simulations of the wind turbine (University of Stuttgart) which are injected in flight-mechanical simulations of a helicopter (provided by Technical University of Munich and DLR Braunschweig). Thus, the tip vortex measurements are an important contribution to the validation of later
5   numerical simulations of the flow.

### 2.3 Available data

For the tip vortex evaluation five flight legs (straight and level fly-by's) are available 0.25 D downstream the WEC rotor plane. Only one of these legs shows the necessary criterion for the circulation and core radius calculation (cf. the following sections). The one leg (two vortex measurements) that fits the criterion will be shown exemplary to present the evaluation method and
10   the analytical solution using the BH model approach, including also the approach by Sørensen et al. (2014). At the remaining measurements it often occurred that the measurement range of the pressure transducer, connected to the 5-hole probe, was

overstepped by the pressure differences created by the blade-tip vortex. These measurements could not be used for evaluation due to the absence of data at the vortex measurement.

The presented data was captured within a 15 minutes lasting flight pattern at about 6.30 PM on a summer day. It can be expected that atmospheric conditions (wind direction and speed, thermal stratification, turbulence intensity) did not change significantly during this period. The average wind speed in the inflow was $8.8 \mathrm{~m~s}^{-1}$ from northerly direction ($5°$) with a turbulence kinetic energy (TKE) of $\approx 0.1 \mathrm{~m}^2 \mathrm{~s}^{-2}$ at hub height. These values have been calculated from a ten second measurement ($\approx 200$ m flight distance) in the undisturbed atmosphere.

## 3  Methods

With the goal to measure detached tip vortices behind a WEC, it is helpful to have at first an understanding of the behaviour of those vortices. Fig. 4 shows the helical vortex pattern forming behind a WEC, by representing the iso-surfaces of the $\lambda_2$-criterion of detached tip vortices from CFD simulation. The fully resolved URANS simulation has been performed by the University of Stuttgart with the compressible flow solver FLOWer (Kroll and Fassbender, 2005), using the Menter SST (Menter, 1994) turbulence model. The modelled rotor is a stand alone generic model of the Enercon E-112 WEC rotor, based on free access airfoil data. For more details regarding the numerical methods, please refer to Cormier et al. (2018) in which the same methods have been applied and described. Figures 5 and 6 give a qualitative impression of the presence of the WEC wake. In both, horizontal wind velocity and turbulence kinetic energy (TKE), the wake and its effects are visible. Farther downstream the helical pattern will start to meander and the symmetrical pattern will dissipate into turbulence. In the near vicinity of the WEC nacelle, these vortices follow a helical pattern. The helical structure is shown simplified by a ring vortex in Fig. 7 which is an approximation of the wake vorticity at high tip-speed ratio. The tangential velocity in this sketch can be split in its horizontal components at hub height (nacelle height). Here the $y$ axis points north (ideally antiparallel to the main wind direction) similar to the conditions at the HeliOW campaign (cf. Fig. 3) and the $x$ axis points east along the UAS flight path. Note that, at hub height, the tip vortex ideally has no $w$ component (Fig. 7) under the vortex ring assumption. Thus, at this height, the tangential velocity can be split into its horizontal components $u$ and $v$. The red rectangle indicates a change of perspective, showing a top view of a vortex spinning in the $x - y$ plane. In reality, from planing flight paths until take-off of the UAS and the actual measurement, the wind direction changes slightly. Therefore, for later evaluations the coordinate system has been rotated into the main wind direction.

### 3.1  Vortex model

To measure and evaluate tip vortices from UAS data an analytical vortex model has to be found. Previous efforts to define a vortex were reviewed e.g. by Jeong and Hussain (1995), comparing several definitions with data from direct numerical simulations and exact solutions of the Navier-Stokes Equations. A universal definition of a vortex or a generally applicable model does not exist. Assuming incompressible flow and an irrotational velocity field, where the curl of the gradient of the velocity is zero, the circulation $\Gamma$, representing the strength of a vortex around a contour $C$, can be connected to the vorticity

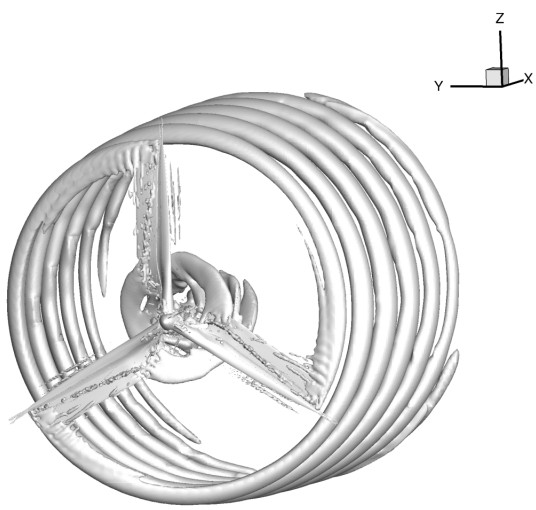

**Figure 4.** Iso-surfaces of detached blade-tip and root vortices following the $\lambda_2$ criterion for vortex identification. Here the $x$ axis follows the main wind direction. Numerical simulation of a generic model of an E-112 4.5 MW converter.

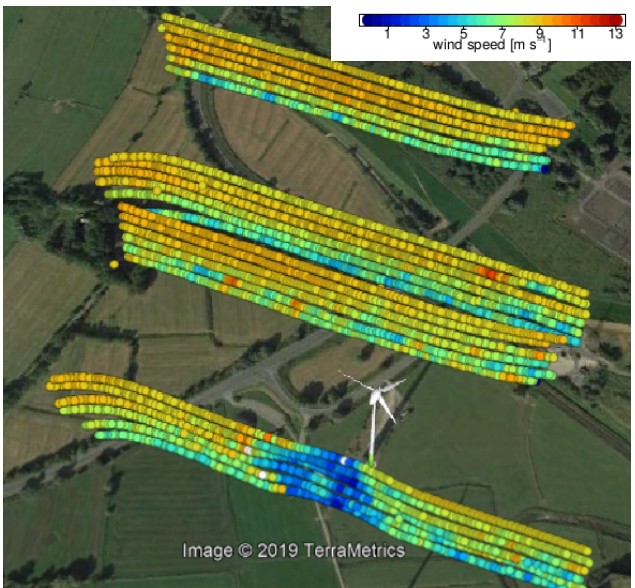

**Figure 5.** Visualisation of the horizontal wind measurements at different flight leg altitudes (from 85 m to 185 m above ground in 20 m steps) and different distances to the WEC (1D, -1D, -2D and -4D). Significant wind deficit 1 D behind the WEC E-112. At this day the wind direction was about 30° north. Image generated in Google Earth.

flux by Stoke's theorem. For any surface $S$ that spans the curve $C$ and $d\boldsymbol{I}$ being an infinitesimal tangential element along $C$,

$$\Gamma = \oint_C \boldsymbol{V_t} \cdot d\boldsymbol{I} = \int_S \boldsymbol{\omega} \cdot \boldsymbol{n} \, \mathrm{d}S. \tag{1}$$

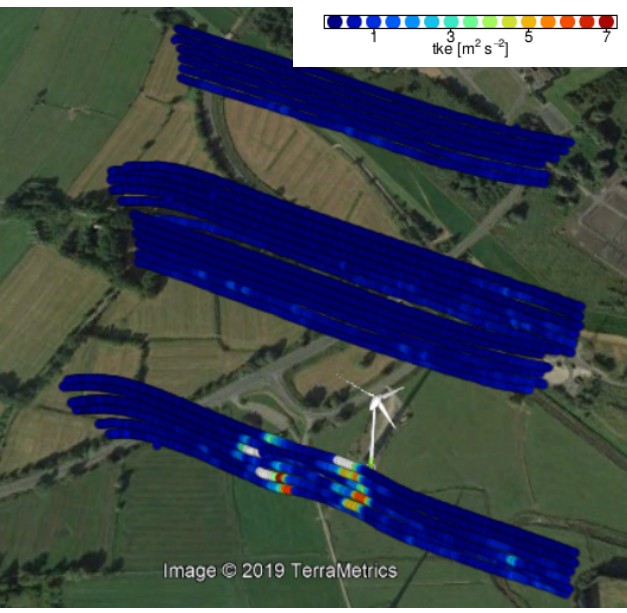

**Figure 6.** Visualisation of the turbulence kinetic energy (TKE) from the same measurements as in Fig. 5. Blue areas represent low turbulence and red the highest measured. At 1D the presence of the wake and the produced turbulence by the tip vortices is visible. The averaging window for the for TKE calculation is 1 s, corresponding to 100 data points, and suits only for qualitative reading only. Image generated in Google Earth.

The circulation $\Gamma$ is the line integral of the tangential velocity along the curve $C$ which is equal to the vorticity flux $\omega = \nabla \times V_t$ through the surface $S$, with $\boldsymbol{n}$ being the normal vector of the surface.

A circular integration in a cylindrical polar coordinate system with the azimuthal angle $\phi$ and the radius $r$ yields:

$$\Gamma(r) = \int_0^{2\pi} \int_0^r \boldsymbol{\omega}(r,\phi) r \, dr \, d\phi \tag{2}$$

5 For a two dimensional, axisymmetric vortex, the circulation

$$\Gamma(r) = 2\pi r V_t(r) \tag{3}$$

is a simple function of the radius and the tangential velocity $V_t$. Since real vortices in fluids experience viscous effects, the structure of detaching tip vortices of the blade of WEC cannot be sufficiently described by Equation 3. Close to the centre of the vortex, lower tangential velocities persist, increasing to their maximum at the core radius $r_c$ of the vortex and decreasing again for farther distances $r$. To account for that, in the context of WEC and also for detaching tip vortices from the wings of aircraft, an analytical model is necessary.

Since in this study detached vortices of a WEC converter are treated similar to aircraft wake vortices, a few similar model approaches were possible. A comparison of analytical vortex models for tip vortices created by aircraft has been done by

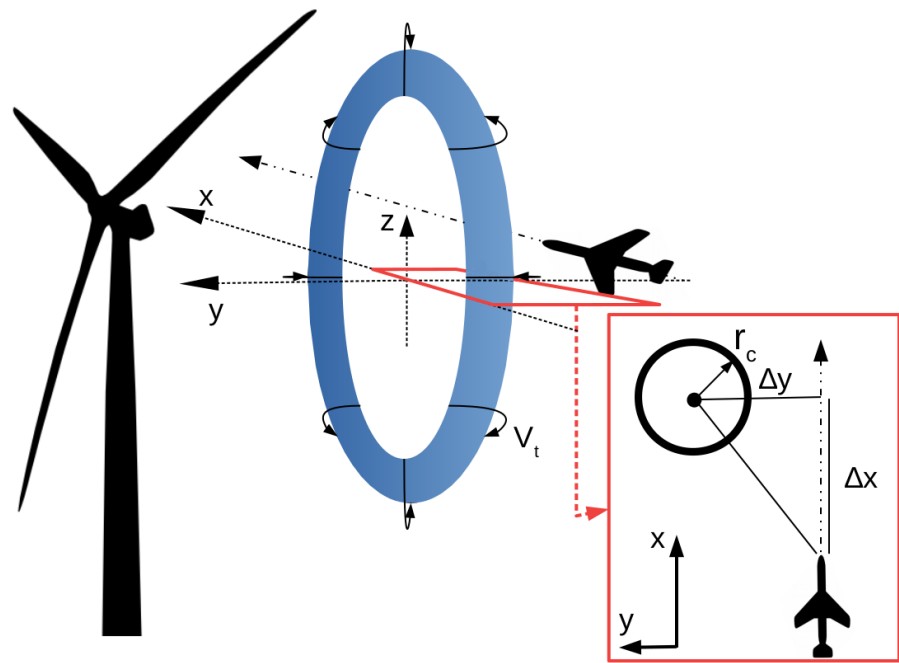

**Figure 7.** Simplified sketch of a vortex pair passed by the UAS to the right. In reality it would rather have a helical pattern than a ring shape. Velocities and axis according to meteorological standards, therefore axis and orientation according to the in-situ conditions. $y$ axis pointing north, $x$ axis pointing east. At hub height the $w$ component (along $z$ axis) vanishes. The red rectangle illustrates a top view of a tip vortex with distance $\Delta y$ to the UAS.

Ahmad et al. (2014). Also Fischenberg (2011) measured wake vortices created by the VFW 614 ATTAS manned aircraft (DLR Braunschweig) and compared the results to two similar vortex models proposed by Lamb (1939) and Burnham and Hallock (1982). Fischenberg concludes that both models show the ageing processes of a vortex wake known from theory. In general the model by Burnham-Hallock shows a slightly better agreement in circulation and tangential velocity to the conducted
5 measurements by Fischenberg. Also Vermeer (1992) uses the BH vortex model successfully to describe WEC wake vortices. According to these findings and its simplicity it has been decided to use the analytical solution for wake vortices by Burnham-Hallock in this study. While the two counter rotating vortices in the BH model used in aviation interact with each other, the two opposite vortices in a WEC wake do not do that. This is an important detail to point out. So for the identification of the vortex parameters ($\Gamma$, $r_c$) a model of two counter-spinning vortices is not necessary. Here, a stand alone vortex is considered. For the
10 later analytical solution of the whole flight path perpendicular to the WEC wake, the BH model for two vortices is consulted.

The BH model does not provide a solution for the whole wake structure, but for an idealised 2D cut. Describing two (independent) counter rotating wake vortices with a simple analytical model and comparing it to in-situ measurements is a new approach in studying wind turbine wake structures.

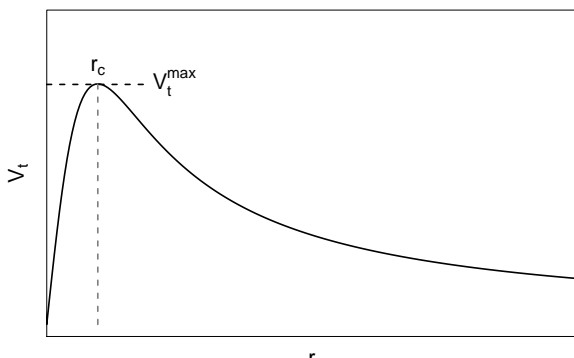

**Figure 8.** Qualitative plot of the tangential velocity from the vortex core outwards. The tangential velocity increases from zero (left) to a maximum at a distance $r_\mathrm{c}$ and decreases to zero for large distances (to the right).

Having a look at the BH model, a vortex is described by its circulation $\Gamma$, tangential velocity $V_\mathrm{t}$ and its core radius $r_\mathrm{c}$. The tangential velocity is the velocity of the air circling the vortex centre and is a function of the distance $r$ to the vortex core.

$$V_\mathrm{t}(r) = \frac{\Gamma}{2\pi} \frac{r}{r_\mathrm{c}^2 + r^2}. \tag{4}$$

The core radius $r_\mathrm{c}$ is defined as the distance from the vortex centre (or core) at which the tangential velocity is at its maximum
(circular symmetry). So the radius $r_\mathrm{c}$ is also the radius at which the surface integral (cf. Eq. 1) is maximal, considering a circular surface. For $r = r_\mathrm{c}$ the maximum tangential velocity becomes (Eq. 5)

$$V_\mathrm{t,\,max} = \frac{\Gamma}{4\pi r_\mathrm{c}}. \tag{5}$$

Figure 8 shows the tangential velocity $V_\mathrm{t}$ distribution of a BH modelled vortex with the highest tangential velocity at the distance $r = r_\mathrm{c}$. The distribution is circle symmetric with the vortex core ($r = 0$) in its centre.
In order to estimate the circulation and size of $r_\mathrm{c}$ from transects through the vortices with MASC in the wake of a WEC, the following procedure is proposed.

## 3.2 Evaluation method

As shown above, it is likely to measure tip vortices at hub height. At this height a simplification of the two vortices can be made. The blade-tip vortices can be considered as two dimensional vortices of circular shape in the horizontal plane and ideally
the $w$ component can be neglected. After subtracting the mean wind $v_\infty$ the vortex tangential velocity is

$$\boldsymbol{v} - \boldsymbol{v_\infty} = \boldsymbol{v'} = (u', v', 0). \tag{6}$$

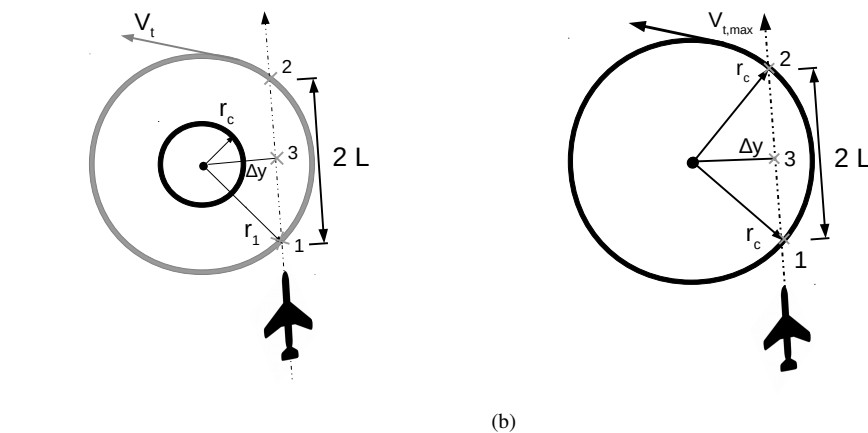

(a)                                    (b)

**Figure 9.** Schematic of the UAS passing a vortex in the horizontal plane (top down view). Two different cases have to be distinguished. The closest distance to the vortex $\Delta y > r_c$ (a) and the passing distance being $\Delta y < r_c$ (b).

The norm of the tangential velocity then is

$$V_t = \sqrt{u'^2 + v'^2}. \tag{7}$$

When measuring with a UAS the measurement can be considered a snap shot of the in-situ conditions. Figure 9 differentiates between two different scenarios of the UAS passing a vortex. Both shown from a top view. Both scenarios will be explained in

detail in the following paragraphs, with first focussing on Fig. 9a. Here the UAS passes the vortex at its closest distance ($\Delta y$), marked point 3 in the sketch, with $\Delta y > r_c$, thus the vortex core radius is not reached. Point 1 and 2 mark the position of two corresponding tangential velocities of identical absolute value, when approaching the vortex and moving away from it again. The measured signal is similar to the dashed black line in Fig. 10 that is an example for $\Delta y = 2r_c$. From such data only point 3 can be identified, since it is the point at which the measured tangential velocity is at its maximum. Point 1 and 2 are somewhere

left and right of the maximum with $L$ being unknown. There are indefinite combinations of $\Gamma$ and $r_c$ that could describe the vortex using Eq. 4.

$$V_{t,2} = V_{t,1} = \frac{\Gamma}{2\pi} \frac{r_1}{r_c^2 + r_1^2} \tag{8}$$

$$V_{t,\Delta y} = \frac{\Gamma}{2\pi} \frac{\Delta y}{r_c^2 + \Delta y^2} \tag{9}$$

$$r_1^2 = r_2^2 = L^2 + \Delta y^2 \text{ (Pythagorean theorem)} \tag{10}$$

The three equations 8, 9, 10 are known to describe the velocities and geometry of the measurement. $V_{t,1}$ ( $V_{t,2}$) is the tangential velocity at the point 1 (and 2). Since there are four unknown parameters $\Gamma$, $r_c$, $L$, and $r_{1,2}$ the problem is not solvable.

Now we consider the case, when the UAS passes a vortex at $\Delta y < r_c$, as shown in Fig. 9b. The measured tangential velocity now provides a distinct feature; a double peak in the horizontal wind measurement. This double peak is caused by passing the maximum tangential velocity at $r = r_c$ at position 1 and 2. Since the tangential velocity decreases from that point inwards

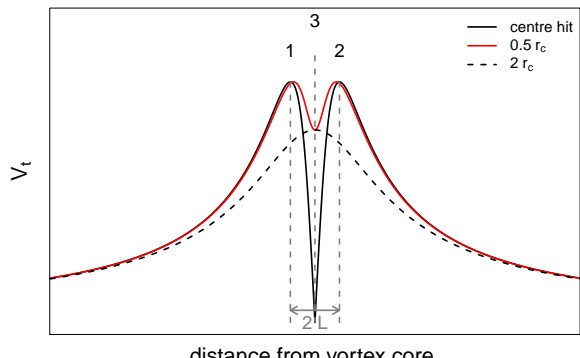

**Figure 10.** Analytical solution of a UAS passing the vortex at a path crossing the centre (black solid line), passing at $r = 0.5 r_c$ (red line) and at a distance double the core radius (black dashed line). The peak to peak distance is $2\,L$ (cf. Fig. 9), above illustrated for the black solid line.

(towards the vortex core), the velocity at point 3 is a local minimum, leading to a visible 'dent' in the data (cf. red line in Fig. 10). Additionally the ground speed of the UAS is known, hence the distance $L$ can be calculated. The three equations previously described above, then become:

$$V_{t,2} = V_{t,1} = V_{t,\mathrm{max}} = \frac{\Gamma}{2\pi}\,\frac{r_c}{r_c^2 + r_c^2} = \frac{\Gamma}{4\pi r_c} \tag{11}$$

$$V_{t,\Delta y} = \frac{\Gamma}{2\pi}\,\frac{\Delta y}{r_c^2 + \Delta y^2} \tag{12}$$

$$r_1^2 = r_2^2 = L^2 + \Delta y^2 = r_c^2 \longleftrightarrow \Delta y^2 = r_c^2 - L^2 \tag{13}$$

With now only 3 ($\Gamma$, $r_c$ and $\Delta y$) unknown parameters it is possible to solve the equations.

Dividing Eq. 12 by Eq. 11 eliminates $\Gamma$. Inserting Eq. 13 gives:

$$\frac{V_{t,\Delta y}}{V_{t,\mathrm{max}}} = \frac{r_c\sqrt{r_c^2 - L^2}}{r_c^2 - \frac{L^2}{2}} \tag{14}$$

Equation 14 describes a tangential velocity ratio that is a function of $L$. Also $L$ is known to range from 0 to $r_c$. A dimensionless relationship $L\,r_c^{-1}$ can be plotted and is shown in Fig. 11. By passing the vortex with $\Delta y < r_c$, and plotting the measured $V_t$ against the distance to the vortex (Fig. 10), we can determine $L, V_{t,\mathrm{max}}, V_t, \Delta y$. Using diagram Fig.11, we finally determine $L\,r_c^{-1}$ and thus $r_c$.

### 3.3  Analytical reconstruction

As shown above, blade tip vortices can be identified by their distinct 'dent' feature, when the $\Delta y < r_c$ criterion is met. Basic geometry and the BH model further allow for a reconstruction (analytical solution) of the individually measured vortex, which

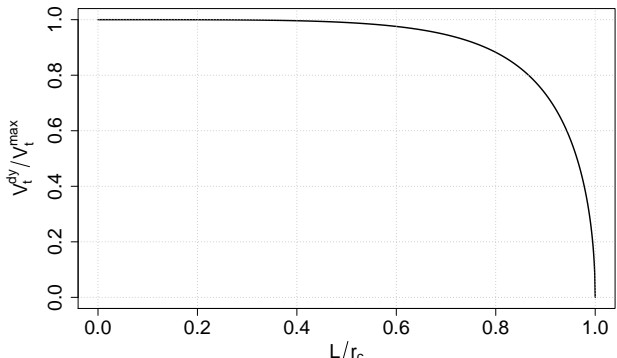

**Figure 11.** Dimensionless relationship between the ratio of the minimum (dent) tangential velocity and the maximum tangential velocity versus half the peak to peak distance ($L$), in percentage of $r_c$.

is helpful to verify the measurements and evaluation technique. With Eq. 15 and 16 the distance to each vortex core (centre), to and along the UAV flight path, can be calculated (Fischenberg, 2011). In Fig. 12 the distances of the UAS to two vortices spinning in opposite direction are shown. In the figure, vortex 1 is passed through its core and vortex 2 is passed with a slight off-set. The flight path of the UAS is indicated with a red dashed line. Those distances are inserted into Eq. 17 and 18 using
5   the relation of Eq. 4 the tangential velocity along the meteorological $x$ axis ($u$' component) and $y$ axis ($v$' component) can be calculated:

$$d_1 = \sqrt{\Delta x^2 + \Delta y^2} = \sqrt{(x - x_{\text{Vortex1}})^2 + (y - y_{\text{Vortex1}})^2} \tag{15}$$

$$d_2 = \sqrt{\Delta x^2 + \Delta y^2} = \sqrt{(x - x_{\text{Vortex2}})^2 + (y - y_{\text{Vortex2}})^2} \tag{16}$$

While the $y$ coordinate can be derived from the measurement (using $\Delta y$ and the UAS position, s.a. chapter 4.3) the $x$
10   coordinate of the vortex $x_{\text{Vortex1,2}}$ is the $x$ coordinate of the flight path at the position '3', e.g. Fig. 9.

$$u' = V_t(d_1)\left(\frac{y - y_{\text{Vortex1}}}{d_1}\right) - V_t(d_2)\left(\frac{y - y_{\text{Vortex2}}}{d_2}\right) \tag{17}$$

$$v' = -V_t(d_1)\left(\frac{x - x_{\text{Vortex1}}}{d_1}\right) + V_t(d_2)\left(\frac{x - x_{\text{Vortex2}}}{d_2}\right) \tag{18}$$

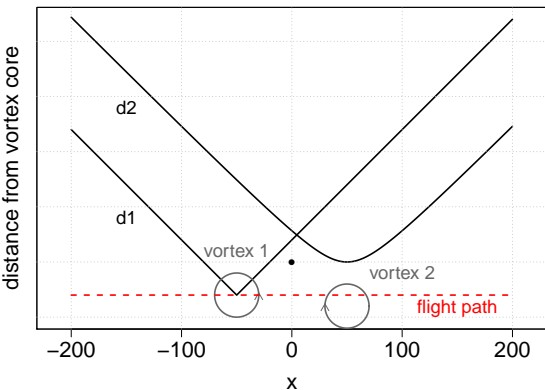

**Figure 12.** Qualitative example of an ideal flight path (vortex 1) and a passing with a little off-set (vortex 2) of the UAS. For the field measurement the distances $d1$ and $d2$ are calculated from the UAS GPS position and the location (off-set) of the vortex in relative coordinates with WEC at $(0,0)$, indicated with a black dot. The vortex position can be derived from the extent of the tangential velocity $V_t$ measured by the UAS and the peak to peak distance, explained in the previous sections. In this example $d_{1,2} = \sqrt{\Delta x^2 + \Delta y^2}$ with $\Delta y_1 = 0$ for $d1$ and $\Delta y_2 = \text{const.} \neq 0$ for $d2$.

## 4   Results

### 4.1   Vortex measurement

Figure 13a shows the $v_h = \sqrt{u^2 + v^2}$ component of the wind measurement behind the WEC at hub height. The data reveals several (near) wake specific features. This flight leg shows two measurements of a tip vortex, indicated by the arrows in Fig.
13a. In-between those two peaks the wake deficit is measurable by a significant drop of the horizontal wind velocity. Due to the near vicinity to the nacelle, the wake deficit is dominated by turbulence created by the blade root vortices. Figure 14 shows a zoomed-in look at the measured vortices depicted in Fig. 13. Figure 14a shows the vortex measured while entering the wake (vortex 1) and 14b while leaving the wake (vortex 2). Both, (a) and (b), are the plain UAS measurements.

Figure 14c and Fig. 14d show the same measurement but the UAS coordinate system is rotated into the vortex rotational
plane. This data rotation is necessary since the UAS travels and measures in the horizontal plane, the vortex rotational plane, however, differs slightly from the horizontal plane. This can be seen in the $w$ component of the plain UAS measurement which is non-zero. To compensate for this fact, for the vortex evaluation, the data are rotated into the vortex rotational plane. This can be understood as the UAS canting into the rotational plane of the vortex to capture the rotational energy in a 2D plane. Through the (individual) transformation an error in the residual time series is introduced, but at a distance of $\pm r_c$ around the
vortex core where the evaluation takes place, the data is corrected. Meaning, in Fig. 14c,d the purple dashed and purple solid line overlay. A good indicator that the data rotation was successful is when the norm of the wind vector (purple dashed line)

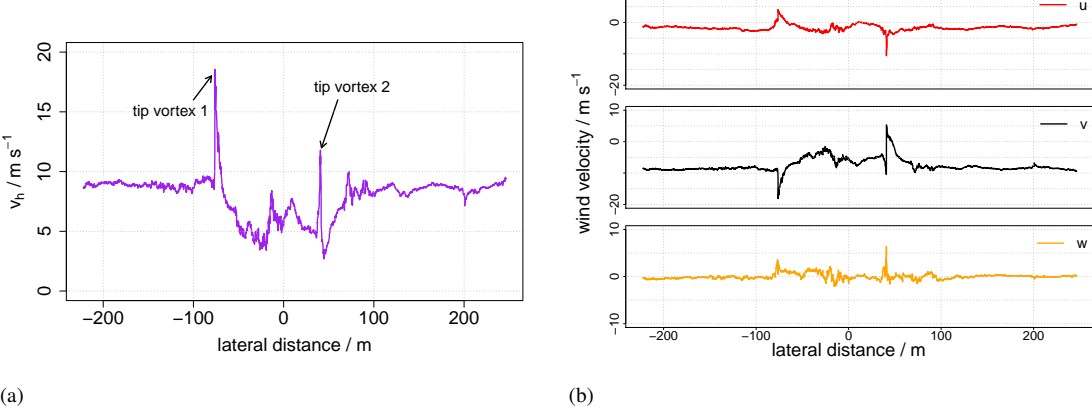

**Figure 13.** (a) Horizontal wind $v_\mathrm{h}$ measurement at hub height in the WEC wake at a distance of 0.25 D to the nacelle. The two tip vortices are indicated by the arrows. (b) The same measurement split into the three wind components $u, v, w$. The $x$-axis is relative easting with the WEC position as origin ($\Delta x$ in Fig. 7).

and the $v_\mathrm{h}$ in-between the grey dashed lines are about the same magnitude. Then it can be concluded that the two dimensional vortex rotation ($u$ and $v$ components) includes the entire kinetic energy, i.e. the vertical wind component is now neglectable.

Examining both vortices, the velocity distribution pattern of the UAS passing at distance $r < r_\mathrm{c}$ is visible in the $v_\mathrm{h}$ measurement. The horizontal wind velocity $v_\mathrm{h}$ is a superposition of the tangential velocity, turbulence and the horizontal wind of the

undisturbed inflow. The characteristics of the tangential velocity of vortex 1 (Fig. 14a,c) is almost solely determined by the $v$ component, while in Fig. 14b,d the $u$ component inheres an equal part. In the plain UAS measurement (i.e. before coordinate transformation) vortex 2 has a significant non-zero $w$ component (Fig. 14b), indicating that the vortex did not rotate in the $x-y$ plane. Especially Fig. 14d shows a significant reduction of the $w$ component after the data rotation. Purple dashed lines indicate the velocity deficit $dV_\mathrm{t}$ (dent), grey dashed lines the peak-to-peak distance. The dot-dashed purple line can be interpreted as an

extension of the horizontal wind velocity by the $w$ component, essentially giving the norm of the wind vector:

$$|\boldsymbol{v}| = \sqrt{u^2 + v^2 + w^2} \tag{19}$$

Table 2 shows the derived parameters from the vortices depicted in Fig. 14. It has to be mentioned that vortex 1 made for a better and clearer measurement, since vortex 2 is influenced by the wind deficit and turbulence inside the wake. Vortex 1 shows a sharp jump in the tangential velocity which makes it easier to obtain the necessary quantities and provides more reliable

results. The average of the obtained circulations is $\overline{\Gamma} = 74.17 \ \mathrm{m^2 \ s^{-1}}$, the average core radius is $\overline{r}_\mathrm{c} = 0.61 \ \mathrm{m}$.

Figure 16 shows a two dimensional cut through a skewed or canted vortex that results in an ellipse where the peak to peak distance is $2\ L'$. This peak to peak distance is under-predicted ($2\ L' < 2\ L$). The introduced error $\Delta y'$ is visualised in Fig. 16 by dotted red lines. To overcome this issue the measured data are rotated into the vortex hose if necessary. This simulates the UAS canting to follow the oblique vortex hose.

**Table 2.** Determined parameters from vortex measurements.

| Vortex | $dV_t$ [m s$^{-1}$] | $V_{t,\max}$ [m s$^{-1}$] | $V_{t,\Delta y}$ [m s$^{-1}$] | $V_{t,\Delta y}/V_{t,\max}$ [−] | $L$ [m] | $L/r_c$ [−] | $r_c$ [m] | $\Gamma$ [m$^2$ s$^{-1}$] |
|---|---|---|---|---|---|---|---|---|
| Vortex 1 | 3.4 | 9.6 | 6.2 | 0.65 | 0.61 | 0.93 | 0.66 | 81.30 |
| Vortex 2 | 0.2 | 9.7 | 9.5 | 0.98 | 0.3 | 0.55 | 0.55 | 67.04 |

## 4.2 Quality control and error estimation

The wake of a WEC, especially as close as 0.25 D behind the nacelle, is a highly turbulent region. When measuring with an autonomous UAS, it is of interest whether the UAS is capable of manoeuvring stably in such an environment and if the measurement instrument (e.g. 5-hole probe) is operating within its operational specifications. Figure 15 shows the attitude of the UAS (a) while passing the WEC for the consulted flight leg and the angle of attack, sideslip and true air speed (b). The UAS is affected by the wake entry and exit. The motions of the UAS are well recognised by the IMU and auto-pilot (cf. Fig. 15a) and taken into account for the later post-processing. The UAS handles these motions without loss of control.

Grey dashed lines in Fig. 15b indicate the limit of the calibrated range of $\pm 20°$ of the 5-hole probe. Passing a tip vortex at $\Delta y < r_c$ is an extreme event, not only for the aircraft, but also for the pressure probe. Angle of attack and sideslip are within the calibrated ranges with one exception of vortex 1. Here, the sideslip is extrapolated. An examination of the true air speed (TAS) for the measurement (blue line in Fig. 15b) shows clearly the entry and exit of the wake. Changes in true air speed cannot be avoided. Usually small deviations from the calibrated TAS value of the 5-hole probe do not result in significant changes in the calculated wind speed. The peaks visible in the TAS measurement however, will have an effect on the wind velocity calculation. The influence of different air speed calibrations on UAS measurements is studied by Rautenberg et al. (2019a). There it is concluded that the deviation from the "true" wind speed is about 10% or at most 1 m s$^{-1}$, e.g. for a TAS error measured at vortex 2 of about 8 m s$^{-1}$. So the peak velocities may be underestimated by 1 m s$^{-1}$.

While this error has no significant influence on the ratio $V_{t,\Delta y}/V_{t,\max}$ it is significant when calculating the circulation $\Gamma$ from Eq. 5. In the presented case the circulation of vortex 2 is under predicted by about 10%.

## 4.3 Vortex reconstruction

The BH model provides a solution for two vortices spinning in opposite direction, as, for example, found in an aircraft wake. A similar constellation of vortex pairs can be found in a WEC wake at hub-height (cf. Fig. 7), with their vortex cores positioned along the $x$ axis. This approximation can only be done, when the flight path is perpendicular to the wind (wake) direction to assure that the measured vortices are of the same age.

With the average values $\overline{\Gamma}$ and $\overline{r}_c$ retrieved from Tab. 2 the minimum measured tangential velocity between the two peaks (position '3' in Fig. 14) as well as a distance $\Delta y$ can be derived. The resulting distance to the vortex core $\Delta y$ can then be fed to a model, based on the BH approach. Figure 17 shows the analytical solution of $u'$ and $v'$ overlain measured data of $u$ an $v$.

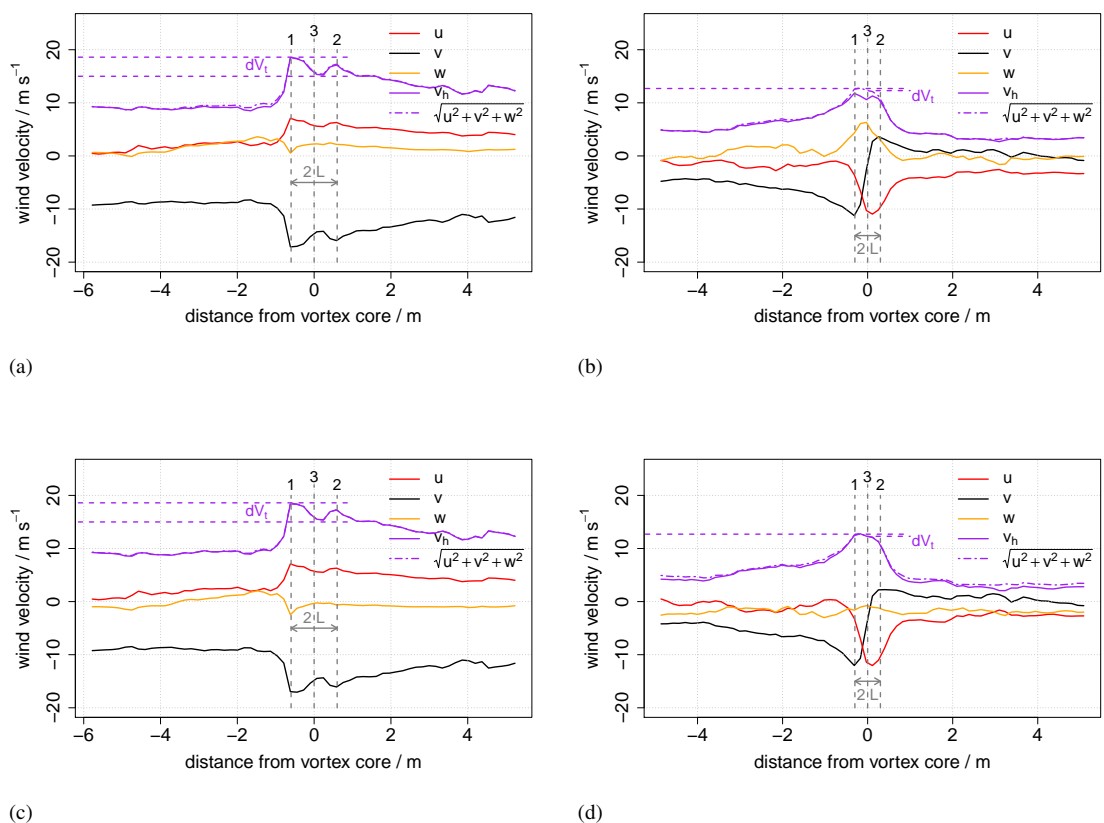

**Figure 14.** Measured tip vortex 1 and tip vortex 2 from Fig. 13a. Purple dashed lines indicate the velocity deficit (dent), grey dashed lines the peak-to-peak distance. The horizontal wind velocity $v_h$ is a superposition of the tangential velocity and the horizontal wind of the inflow/surroundings. To eliminate the $w$ component the data has been rotated into the vortex coordinate system. This is necessary to measure the vortex correctly. The sub-figures (a) and (b) show the plain UAS measurement of the vortices. In sub-figures (c) and (d) the UAS has been rotated into the vortex coordinate system (vortex plane) to capture the whole two dimensional rotation.

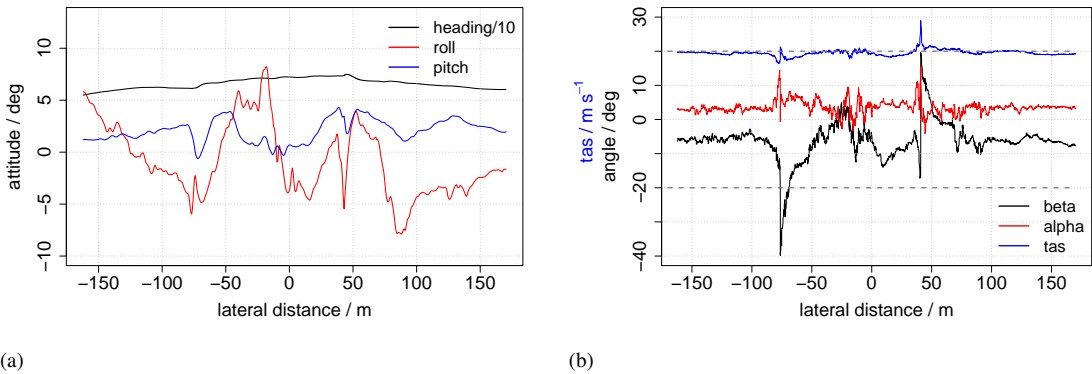

**Figure 15.** (a) Attitude angles of the UAS from flight leg taken for the tip vortex evaluation. (b) Angle of attack (alpha) and sideslip ($\beta$) at the 5-hole probe. Grey dashed lines indicate the calibration range of the 5-hole probe. Overstepped angles are extrapolated in post-processing. In blue the true air speed of the UAS.

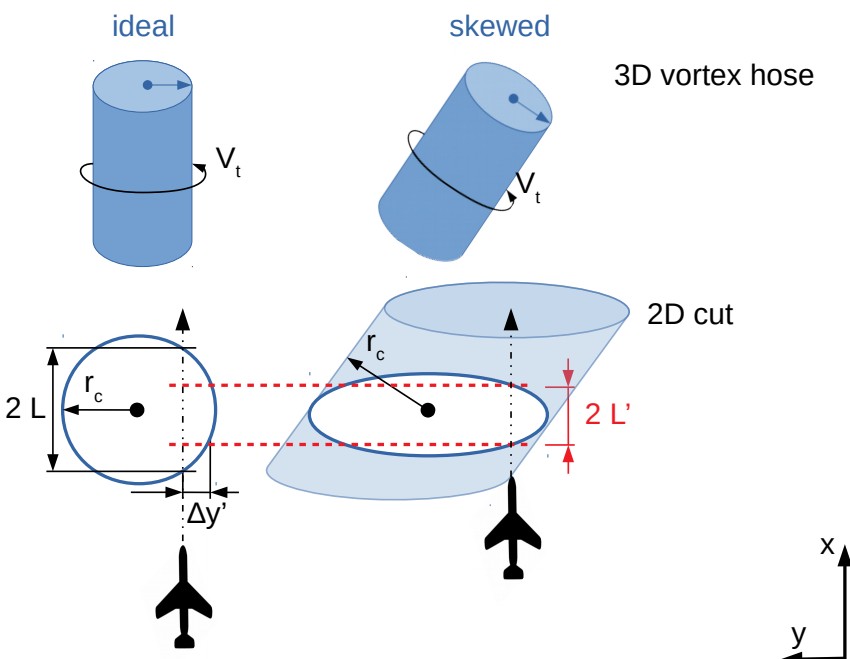

**Figure 16.** Sketch of an ideal and skewed (exaggerated) vortex hose at hub height. The simplifications in the evaluation method only consider components in the $x-y$ plane which leads to an under-prediction of the real peak to peak distance. The fact that the real '$r_c$' does not lie in the $x-y$ plane leads to an error. A horizontal cut though the vortex has an ellipsoidal geometry instead of a circular one, as in ideal measurement conditions.

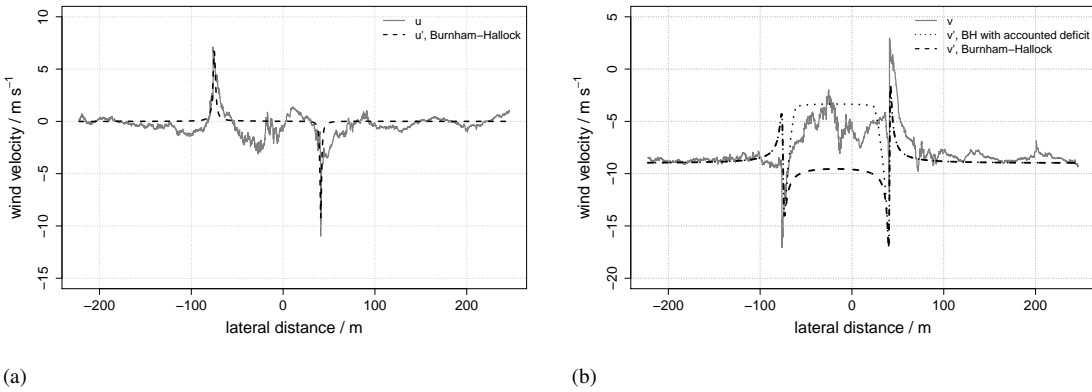

**Figure 17.** Analytical solution (dashed lines) for $u'$ and $v'$ of the two vortices from the parameter evaluation. The corresponding wind components $u$ and $v$ from (Fig. 13b) from the UAS measurements in grey. The long dashed solution in (b) additionally accounts for the deficit in the wake.

Overlain to the in-situ data the tangential velocity still contains the mean horizontal wind $V_\mathrm{t} = \sqrt{u'^2 + v'^2}$. For the analytical solution the measured data has been rotated slightly (ca. $10°$) into the mean wind direction to fit the meteorological coordinate system with the vortex coordinate system, so the $u$ component equals zero in average and $v$ is the predominant horizontal wind direction. In addition to the solely BH solution for the $v'$ component (dotted line in Fig. 17b), the long dashed line shows the

same solution, but multiplied with a correction factor to satisfy for the wind deficit in the wake. The general vortex model does not consider the mean horizontal velocity, so it needs to be accounted for, especially when there is a artificially induced drop behind the WEC in the wake (wake deficit). In the present case the velocity deficit was measured to be about 65 %. It is visible as a jump in the mean horizontal wind between the two measured vortices. Similar deficits were already measured by Wildmann et al. (2014a) or Bartl et al. (2012). The velocity correction function is simply an upside down Tukey window.

The analytical solution remains uncorrected until entering the wake of the WEC. After incorporating a deficit correction to the analytical solution, it is visible that the deficit in the wake plays an important role to the structure (placement, intensity, etc.) of the vortex. Especially since the two vortices do not interact with each other, as the two vortices in the BH model for aircraft wakes do.

## 5   Discussion

Here we compare the airborne measured circulation $\Gamma$ with data of the WEC itself. Equation 20 allows for a calculation of the blade-tip vortex strength by given parameters and describes the circulation for a rotor of constant thrust coefficient, e.g. (Sørensen et al., 2014):

$$\Gamma = \frac{\pi v_\infty^2 C_T}{\Omega N_\mathrm{b}} \approx 66.2 \ \mathrm{m}^2 \ \mathrm{s} \tag{20}$$

$N_b$ being the number of blades and $\Omega$ the rotational velocity provided by the owner of the WEC. For the determination of the thrust coefficient $C_T$ the following estimation is done:

The relatively low wind speed ($v_\infty = 8.8 \text{ m s}^{-1}$ by UAS measurement) implies a pitch angle of $\beta = 0°$ when approximating the E-112 with the NREL 5 MW offshore WEC (Jonkman et al., 2009).

5      The tip-speed ratio ($TSR = \frac{\Omega R}{v_\infty}$) can also be calculated and thus a thrust coefficient $C_T \approx 0.8$ can be estimated from the $C_T$ to $TSR$ relationship by Al-Solihat and Nahon (2018).

The calculated value for $\Gamma$ from WEC specific and atmospheric parameters is similar to the vortex strength that was extracted from the vortex measurements (average $\overline{\Gamma} = 74.17 \text{ m}^2 \text{ s}^{-1}$). The presented method, to calculate Gamma from UAS data, using a geometric simplification of the tip vortex and the application of the BH vortex model, provides reasonable results.

10      The BH vortex model does work for aircraft induced vortices as shown by Ahmad et al. (2014) as well as Fischenberg (2011) and as the results imply, it can be used to describe WEC wake vortex properties. Not least, both phenomena can be described by two vortices spinning in opposite direction, yet there is no interaction of the two opposite vortices, as in the aircraft wake model usually intended. Vortex patterns of a WEC wake show higher complexity than aircraft wake vortices. The whole wake is in motion and different turbulence and shear forces interact with each other. Therefore, for the wake vortices some simplifications 15 had to be made, e.g. the shown evaluation method is only valid for a 2-D cut of the whole vortex hose. Also the blade root vortex was not analysed any further.

In this study also the fact that the UAS experiences a change in true air speed (TAS) when entering the wake is addressed. Theoretically the calibration range of the used five hole probe is for a fixed air speed which changes when entering the wake. Since this evaluation uses the ratio of two velocities the influence of a different calibration for the five hole probe does not lead 20 to a significant error. For the calculation of the circulation $\Gamma$, however, absolute velocities are necessary and a small error can be expected due to a change in TAS when entering and leaving the wake velocity deficit. The error is estimated to be $\pm 10\ \%$ for the calculated wind velocities (Rautenberg et al., 2019a). An error estimation is given in Section 4.2.

## 6   Conclusions and outlook

The resulting circulation strength $\Gamma$ derived from UAS data shows good accordance to the results obtained from Eq. 20. It can be 25 concluded that the evaluation method, using the basic geometrical properties of a vortex, can be used to derive vortex properties in a WEC wake. Turbulence acting on the vortex and on the surrounding atmospheric flow can aggravate an evaluation since the evaluation is done mainly graphically. For example the second tip vortex is embedded in a relatively high level of turbulence (wake deficit, shear, etc.). It also does not show a clear border to the undisturbed atmosphere as tip vortex 1 does. The reference velocity levels for the evaluation are therefore harder to extract from the measurements. Also a hit of a blade-tip vortex in flight 30 changes the true air speed (TAS) locally and temporally, resulting in an error in the velocity measurement (usually $5 - 10\%$ off).

In addition, this method still has to be proven at larger distances to the WEC nacelle, where the vortices might begin to meander and get unstable. However, to our knowledge, this is the first quantitative analysis of WEC tip vortices using in situ measured turbulence data by a fixed wing UAS.

The MASC Mk 3 system is capable of measuring detached tip vortices in the wake of a WEC. The spatial and temporal resolution is sufficient to detect vortex patterns in the measurements. However, on many occurrences, the measured sideslip $\beta$ left the calibration range of the 5-hole probe in a matter that the corresponding pressure transducer was off the measuring range, leaving data lags in the time series. In conclusion, those measurements could not been used.

For future measurements the calibration of the (conical) 5-hole probes could simply be expanded to larger angles up to $\pm 40°$ (Fingersh and Robinson, 1998). This then allows for a lower TAS (true air speed) of the UAV, which in turn results in lower pressures at the pressure transducers and a better spacial resolution of the data. The path accuracy of the UAS will be upped by using an RTK (real time kinematic) GPS. This will allow for precise back-calculations of the positions of the vortices. Wake meandering, wake and vortex widening can then be documented.

The proposed analytical vortex model by Burnham and Hallock is capable of describing WEC wake vortices. Yet, as for most analytical models, the analytical solution shown in this paper can and should be improved. E.g. to better fit the WEC wake (velocity deficit, blade root vortex near the nacelle). This evaluation was conducted with data obtained at 0.25 D from the nacelle. For a future additional field campaign blade-tip vortices in the farther wake shall be investigated.

*Competing interests.* The author declares that he has no competing interests.

*Acknowledgements.* We acknowledge support by Projektträger Jülich and the BMWi (Federal Ministry for Economic Affairs and Energy) that funded the HeliOW project. We thank Enercon GmbH for cooperation and WRD GmbH for the provision of a generic recreation of the E-112 geometry for the numerical simulations. We also acknowledge support by Deutsche Forschungsgemeinschaft and Open Access Publishing Fund of University of Tübingen. For extensive technical support at the field campaign we want to thank Martin Schön and Patrick Manz.

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
