# Peer review of "First Identification and Quantification of Detached Tip Vortices Behind a WEC Using Fixed Wing UAS"

_Wind Energy Science, 2019_

## Short Comment (SC1) · 19 Mar 2019

The following publication deals with the issue of airspeed variations for measurements with MASC-3. By analysing the influence of the calibration of the five-hole probe, the expected uncertainty when transecting the wake of a wind turbine and a tip vortex are presented.

Title: Calibration Procedure and Accuracy of Wind and Turbulence Measurements with Five-Hole Probes on Fixed-Wing Unmanned Aircraft in the Atmospheric Boundary Layer and Wind Turbine Wakes

Reference: Atmosphere 2019, 10(3), 124; https://doi.org/10.3390/atmos10030124

The error is estimated according to Rautenberg et al. to be $\pm 10$ % for the calculated

wind velocities. Applied to Eq. 20 in Section 5 of the discussion paper, the calculated circulation strength $\Gamma$ may range between $54 - 80$ m$^2$ s$^{-1}$.

---

## Author Comment (AC1) · 20 Mar 2019

Those values for the error of the circulation strength are based on the assumption that $U_\infty$ is the faulty variable. But that is not the case. The faulty variable is the circulation $\Gamma$. So there is an error in the evaluation of the peak velocity of the tip vortex (1 and 2) but not in the calculation of the 'free stream velocity' (horizontal wind outside the wake).

Additionally one has to look at the TAS (true air speed) at the time of measuring the tip vortex. For the tip vortex 1 the TAS is calculated to 17.2 m/s, that is 0.8 m/s off the calibration velocity. So this vortex evaluation should be fine. Hence the TAS at the tip vortex 2 (leaving the wake again) is calculated to 14.2 m/s which is about 3 m/s off. So

there is a distinct, small error in the calculation of the peak velocity there. This error is introduces due to the wind velocity drop in the wake deficit. This error in tip vortex 2 affects the determination or $r_c$ and L. And later the circulation calculation.

To sum it up. There is an influence of the calibration (TAS drop) but it is nowhere near these $30\ m^2/s$.

---

## Referee Comment (RC1) · Anonymous Referee #1 · 25 Mar 2019

**1   General comments**

The manuscript by Mauz et al. describes in-situ measurements of wind turbine tip vortices with a UAS. From these measurements, circulation of the vortices is calculated using the Burnham-Hallock wake vortex model. These measurements are unique and I do not know of any other study in which a UAS operated in such close proximity to an operating wind turbine and even in its wake. The authors can convincingly show that wake vortices can be measured with the UAS. The presentation of the methods of analysis and the results however needs some significant improvement:

- In the introduction and throughout the text I miss more thorough references to the

state-of-the-art. For example, other methods to measure full-scale wind turbine wakes with remote sensing are not mentioned at all, but are carried out all the time and in multiple ways. What can a UAS do that is not possible with remote sensing?

- The structure of the text is sometimes confusing and much information is given in the results and discussion and outlook section that should have been introduced before. I provide details in the specific comments.

- A major problem of the manusript is that all the analysis is only done for a single sample of a wake vortex pair. If possible at all I would strongly urge the authors to investigate if they can maybe use some other flights. Maybe even flight levels above or below hub height could still be valuable.

- It is known that the estimation of the UAV attitude is a major source of error for the wind calulation. It is also known that navigation systems are less precise in dynamic flight maneuvers. I would therefor at least expect that attitude angles as well as airspeed and flow angles during the flight through the vortex are shown. The authors raise the issues themselve in the discussion, but I think it is necessary to include a proper analysis of this in the manuscript, including an estimation of uncertainty of the wind vector and thus the circulation.

- I recommend some copy-editing to be done on English grammar and expressions.

For these reasons, I suggest a major revision of the manuscript before considering it for publication in WES.

**2 Specific comments**

**2.1 Abstract**

The abstract should be rewritten with more statements about the hypotheses that were investigated in the study. A description of the results / conclusion is missing.
*p.1,ll.1ff*: wind converter should be either wind turbine or wind energy converter. Preferrable: "rotor blade of a wind turbine". The whole first sentence is very hard to read and grammatically wrong. Starting with the relevance for numerical models in the abstract is misleading, because numerical models are not the topic of the paper.
*p.1,l.6*: what is the difference between atmospheric and meteorological quantities?

**2.2 Introduction**

*p.2,ll.9-13*: this paragraph should go into the description of the aircraft (Sect. 2.1)
*p.2,ll.14-21*: this paragraph could go in the experiment/site-description in Section 2
*p.2,ll.22-31*: here, a lot more literature should be referenced. Wake vortices are a major field of research and the state-of-the-art has not been evaluated at all

**2.3 Section 2**

- I miss a detailed description of the atmospheric conditions during the flight. Stratification, turbulence in the inflow, etc. which are important for the wake development are not given but should be available from the data.

- A list of available flights and an explanation why only a single flight leg is analysed is missing.

*p.3,ll.3-8*: Here, description of the aircraft is mixed with operational procedures. I feel like this should be separated.

*p.3,l.11*: Since RTK GPS is mentioned here and is not a standard feature of UAS in atmospheric research, some additionoal information would be appreciated: what kind of receiver is used (L1 or L1/L2 phase); is a local base station or RTCM-services used for correction data / what is the baseline? What is the advantage of the very accurate flight path in atmospheric measurements?

*Fig.3*: I think a more schematic background (not Google Earth) with a better scale and legend would help

**2.4   Section 3**

*Fig.6*: It is unclear how TKE has been calculated. How large is the averaging window?
*Fig.7*: Nice figure, but watch out for which lines cross in front of or behind other lines to get the 3D visualization right. I think the red rectangle encloses the blue vortex, right? And "distance" should probably get a variable name or could even be left out
*p.9,l.5*: Except that the vortices along the horizontal axis are not generated at the same time for the WEC.

**2.5   Section 4**

*p.14,l.1*: What is the vortex coordinate system and which angles are used for the rotation? This has not been introduced before.
*p.14, ll.26ff*: this needs to be introduced and explained in more detailed in the methods section. Why can this correction not be done for other flight levels to increase the number of samples of tip vortices?
*Fig.16*: I think this figure is not necessary
*p.15, l.5*: "rotated slightly" -> what does that mean?

*p.15, l.10ff*: It is said that the wind speed deficit plays an important role for the tip vortex, but this is not discussed any further. Is BH even an appropriate model under these circumstances?
*Fig.18*: not sure if this figure is necessary

**2.6 Section 5**

*p.19,l.7f*: These are too strong statements for an experiment with a single sample
*p.19,l.15ff*: The issues that are raised here are not insignificant and call for some more analysis and quantification.

**2.7 Section 6**

*p.19,l.21*: What is the equation by Sorensen et al. (2014) that is meant here?
*p.19,l.23*: "aggravates an evaluation" - what do you mean by that?
*p.19,ll.27ff*: The information about the 5-hole probe calibration range and why other flights could not be used should be given in Section 2 already. In section 2, it was said that the UAV operated with RTK GPS which is contradicted here.

**3 Technical corrections**

*p.1,l.1ff*: wind converter should be either wind turbine or wind energy converter. Preferrable: "rotor blade of a wind turbine". The whole first sentence is very hard to read and grammatically wrong. The second sentence raises problems of numerical simulations that are not the topic of the paper.
*p.1,l.6*: what is the difference between atmospheric and meteorological quantities?
*p.1,l.8*: $u,v,w$, italic please

*p.1,l.18*: "their individual capacity and diversity" (what do you mean by diversity?)

*p.2,l.3*: underestimate

*p.2,l.4*: "Another way" or "Another method"

*p.8,l.7*: "several analytical models are available"!?

*Fig.8*: x-axis should be labelled $r$

*p.11,l.18*: $L$ is not proportional to the velocity ratio -> The velocity ratio is a function of $L$.

*p.14,l.10f*: "In Fig.14 shown as a solid purple line." What is?

*p.14,l.21*: "clear and sharp jump" - strange expression

*p.14,l.22*: Fig. 16 appears before Fig. 15 in the text.

*Fig.17*: What is $u$ and what is $v$ should be mentioned in the caption. the same line style should be used for the same analysis method, i.e. dashed line for simple BH, and dotted for corrected version for example.

---

## Referee Comment (RC2) · Anonymous Referee #2 · 7 Apr 2019

I agree with the comments from RC1, so I will focus on some of the other things I noted. I have written down some general comments below, followed by a list of specific comments.

General comments:

As it is, I found the article hard to follow, which is unfortunate since the results are interesting. The authors should focus on guiding the reader through their thoughts and results, in a clear and concise way, reducing the length of the paper. This will require a major revision of the paper.

I also believe the authors should spend an important amount on time correcting the language of the article. I've highlighted a couple of paragraph and sentences that

could be improved up to page 11, after which I stopped mentioning these. The authors should still work on the text after page 11.

Regarding the title, I would probably not support using the word "first" in the title since it brings a competitive touch to it that is unnecessary in my opinion. Also in light of the following publication, it may unfortunately not be justified to claim this "first" attribute: F. Carbajo Fuertes et al - 2019 - "Multirotor UAV based platform for the measurement of atmospheric turbulence: validation and signature detection of tip vortices of wind turbine blades.". The author may also consider the studies from, Kocer et al. 2011 and Reuder and Jonassen 2012 cited in the above reference. (Please note that I'm not an author of any of these papers). Yet, I leave this up to the authors and the editor to decide whether to change the title.

I would personally prefer the equations to be closer to the text. As it it now, the equations are usually floating at the end of the paragraph which can make the discussions hard to follow.

It appears that the method presented can be attributed to Fischenberg, and the author may need to be clearer when highlighting if something is new or unique in their approach compared to what was already published (apart from the measurement campaign). The model using a regularized vortex cannot really be seen as a new contribution or method. The experimental data though are of high value.

The amount of measurements appear unclear, and some statistical analysis and information about the ensemble of results available could be valuable. Reading the article, it seems that only one vortex was analysed. Further data with different distances downstream should be incorporated since according to section 2 more data was acquired. Statistical tools should also be used to mention the uncertainty on the fitted parameters and to quantify the error between the model and the measurements.

The figures are usually clear. The authors could yet reduce the number of figures, particularly in the first 10 pages, or by combining the measurements with the fitted

model in figures 10-17.

I hope my numerous comments will not discourage the authors, and I strongly encourage them to further work on this paper. As I mentioned earlier, the article has some great potential, it just needs some additional work to reach the point of publication.

Below is a list of more specific comments:

Astract: The statement in the abstract "the BH model can be used to describe wake vorticies" is probably too strong and would need to be moderated since this simpler model is not capturing all the dynamics. I will comment more on this later.

Introduction: I would think that bringing the context of Germany appears too specific, since the wind energy sector is growing in other countries.

p1 l21: "In research..." this sentence and the following two are hard to read and could be reformulated

p2 l8: The scaling problem of wind tunnel measurements could be mentioned here

p2 l10: make sure the acronym for UAS (and other acronyms) is made explicit in the introduction

p2 l15: Could you mention the arguments for the offshore comparison. It probably relies on arguments on the boundary layer when the flow comes from the shore, but wave exciting the turbines and the surface roughness may be different.

p2 l18: "The project aims for save helicopter flight paths in off-shore wind energy parks " needs reformulation

p2 l19, l21 *University of* Munich, University *of* Stuttgart.

p2 l23: the wind turbine also generates strong coherent vorticies, can these be considered turbulence?

p2 l30: could you highlight more the difference between the study from Subramanian

and yours?

p3 l7: Wind speed and directions could have changed during this 15min period, do you have access to measurements to support this assumption?

p5 l11: You could mention that the ring vortex is an approximation of the wake vorticity at high tip-speed ratio.

p5 l14: This line can be reformulated to mention that this result is true under the vortex ring assumption.

p6 Figure 4: "recreation" may not be the correct word in the caption, maybe "model", or "reproduction" would be more accurate?

Figure 7: Instead of using north/east for the axis wouldn't it be easier in that case to use an orientation in the frame of the turbine, with y pointing upstream against the main wind direction? You could then remove the sentence at the end of page 6.

p7 line1-12: The potential flow assumption probably appears too early in this paragraph and the paragraph could be reformulated. The definition of circulation as function of the vorticity is independent of this assumption. Equation (3) only uses an axi-symmetric assumption. It is yet true that the circulation of a vortex makes more sense in inviscid flows where the vorticity is condensed to confined singular regions.

p10 l1-4: This paragraph needs should be reformulated, the language improved.

p10 l9-15 and p11 l1-8: While reading the text I was confused since figure 9b didn't appear to be mentioned. The explanation could be improved by clearly explaining both figures and both scenario, before mentioning figure 10. Alternatively, you could tell the reader to focus only on figure 9a for now and figure 9b will be explained later. Also, equations 8-10 could be introduced first before drawing the conclusions that there is no unique solution. Further, the way the equations are introduced can be improved by telling to the reader what is coming, e.g. "The velocities at point 1 and 2 are...". Right now, they appear in the text announced.

p11 l9: " This double peak can be lead back to passing the maximum tangential velocity at r = r c at position 1 and 2". The sentence may need reformulation

p11 l12-15: Similar to the previous remark, the equations can be introduced before drawing the conclusions.

p11: "As shown above the presence and identification of a vortex (or a pair of vortices) is measurable". The language needs to be reformulated (one cannot measure an identification). Also, the previous section seemed to show that in some cases the determination was not possible, which would imply that the identification is not always possible.

p13 l8: It would help the reader to provide some information about the measurement campaigns (how many samples were selected, what were the mean conditions), and some introduction about the samples you selected (and why you selected them) to present in the paper.

p14 l26: It is not clear why you mention the skewed vortex at this stage. You may need to guide the reader. Also, the definition of the skewed vortex on figure 15 appears unclear. From the figure it seems that the vortex is simply rotated. I'm not sure this qualifies as a skewed vortex. Also the 2D cut appears to have some 3D aspect to it, which can be confusing. Introducing a coordinate system on the 3D vortex on the 2D cut can help the understanding.

p14 l30: I am wondering if "analytical reconstruction" is the proper term and how this "reconstruction" is different from the previous section. The parameters you extracted from the measurements were fitted to an analytical model. The "reconstructed" vortex is this analytically modeled vortex, and it is intrinsically part of the results you presented in table 2. If I understand this correctly, it could make sense to have the modelled vortex directly on the figures 13-15, as "fitted vortices".

p15 l9: What is meant by "artificially induced drop" of velocity? Does this refer to drop

of velocity in the turbine wake? If this is so, the drop in velocity should be a function of the thrust coefficient at the rotor, and a value of 65% may not be comparable to other measurements unless they are at the same operating conditions.

p18 eq20: You may have to introduce all symbols closer to the equation, even if these are obvious.

p19 l1-4: It appears suprising that the authors do not have more information about the turbine (thurst curve, pitch curve). Earlier in the text, it was mentioned that a model of the turbine was done. These quantities can then be obtained from a Blade Element Momentum code.

The argument here may simply be that most turbine have a pitch angle around +/- 1 degree below rated, and in the absence of data, you picked 0 degree. It is also not clear where the pitch angle enters in the equation. Most likely an argument of the thrust coefficient, but usually you'll have a thrust coeffcient vs wind speed curve available for that turbine.

p19 l10-11: These sentences needs to be moderated. First, it appears that the study was only done on one vortex at a given operating conditions and a more quantitative analysis would be required. Second, it appear wrong to state that the wake of a turbine is described by two vorticies. You could clarify your discussions based on the following considerations. The fact that two vortices are crossed on the trajectory of the drone is due to the likelyhood of crossing the tip-vortices from different blades. This likelyhood increases as the number of blades or the tip-speed ratio increases. When it is such, the wake vorticity surface can be approximated to a vortex cylinder, in which case any trajectory of the drone will indeed cross the tip-vorticity surface twice. This cylindrical surface does not ressemble two vorticies spinning in opposite direction and the wake dynamcs cannot be described by assuming that it consists of two vorticies. What the author probably mean is that the velocity field accross a tip-vortex (or a tip-vorticity surface) ressembles the one of a regularized point vortex. This analogy (which is natural

given the different analytical vortex wake models of wind turbines) cannot be used to "describe" the wake, but it can be used to "estimate" some of the wake properties, that is, the tip-vortex core radius and circulation.

p19 l21: The identification of one vortex strength do not appear to be enough to draw a conclusion, or the concluson needs to be moderated. Also, it may not be necessary to attribute this equation to Sorensen et al and instead it can be mentioned where this formula comes from: circulation for a rotor of constant thrust coefficient (This should also be mentioned earlier in the text p18 l1-6).

---

## Author Comment (AC2) · 24 Apr 2019

**Author's response to Referee #2**

April 24, 2019

Thank you for the detailed review of the manuscript. In the following I will comment on each point. The referee's comments will be repeated in italic before the answer.

**General comments**

*I agree with the comments from RC1, so I will focus on some of the other things I noted. I have written down some general comments below, followed by a list of specific comments. As it is, I found the article hard to follow, which is unfortunate since the results are interesting. The authors should focus on guiding the reader through their thoughts and results, in a clear and concise way, reducing the length of the paper. This will require a major revision of the paper. I also believe the authors should spend an important amount on time correcting the language of the article. I've highlighted a couple of paragraph and sentences that*

- *I also believe the authors should spend an important amount on time correcting*

*the language of the article. I've highlighted a couple of paragraph and sentences that could be improved up to page 11, after which I stopped mentioning these. The authors should still work on the text after page 11.*
I will look into that.

- *Regarding the title, I would probably not support using the word 'first' in the title since it brings a competitive touch to it that is unnecessary in my opinion. Also in light of the following publication, it may unfortunately not be justified to claim this 'first' attribute: F. Carbajo Fuertes et al - 2019 - 'Multirotor UAV based platform for the measurement of atmospheric turbulence: validation and signature detect ion of tip vortices of wind turbine blades.'. The author may also consider the studies from, Kocer et al. 2011 and Reuder and Jonassen 2012 cited in the above reference. (Please note that I'm not an author of any of these papers). Yet, I leave this up to the authors and the editor to decide whether to change the title.*
Thank you for your valuable input there. The paper by Carbajo Fuertes was put online on March 28 2019. So there was no possibility to the authors to have known from these results.
Following your suggestion I had a look at Fuertes 2019. The results look good. It is nice to see that two different approaches and measurement systems come to a similar result and output. Fuertes also derives a circulation of $50\ m^2/s$ from his measurements. Given that he measured at a different wind turbine, the circulation strength is of the same order of magnitude. However, I could not find any information about the vortex fit he is using for his measurements as well as a detailed information about the circulation calculation.

- *I would personally prefer the equations to be closer to the text. As it is now, the equations are usually floating at the end of the paragraph which can make the*

*discussions hard to follow.*
I will consider this point to be mainly to be an editorial problem. I will see, to put
equations closer to the text. However, in the later two-column layout everything
will change again. So I might need input from the editor at this point.

- *It appears that the method presented can be attributed to Fischenberg, and the
author may need to be clearer when highlighting if something is new or unique
in their approach compared to what was already published (apart from the mea-
surement campaign). The model using a regularized vortex cannot really be seen
as a new contribution or method. The experimental data though are of high value.*
It is correct that the method and model is not new. But as I state that the model
has its origin in aviation, it is new to be used to describe wind turbine wake vor-
tices properties. I added a paragraph to highlight this new approach in WEC
wake studying.
Also the described method illustrates how to obtain results from in-situ line
measurements of a UAS, including systematic measurement uncertainties like
skewed/canted vortex hoses etc.

- *The amount of measurements appear unclear, and some statistical analysis and
information about the ensemble of results available could be valuable. Reading
the article, it seems that only one vortex was analysed. Further data with dif-
ferent distances downstream should be incorporated since according to section
2 more data was acquired. Statistical tools should also be used to mention the
uncertainty on the fitted parameters and to quantify the error between the model
and the measurements.*
A new subsection '2.3 Available data' has been added. I also state why only one
pair of vortices could be used for an evaluation. A simple vortex measurement

in the wake is not sufficient, also the criterion $\Delta y < r_c$ must be met which is a rare condition. I comment on this in the 'specific comment' section below in more detail.

- *The figures are usually clear. The authors could yet reduce the number of figures, particularly in the first 10 pages, or by combining the measurements with the fitted model in figures 10-17.*
  I will take out some figures from the manuscript. But also some new ones will be added. Mainly to do the uncertainty evaluation (wind angles, true air speed, etc.). I will also try if some figures could be combined. But that might add to the readers confusion.

- *I hope my numerous comments will not discourage the authors, and I strongly encourage them to further work on this paper. As I mentioned earlier, the article has some great potential, it just needs some additional work to reach the point of publication.*
  Highly appreciated emphasis. Thank you!

**Specific comments**

Abstract

1. *The statement in the abstract 'the BH model can be used to describe wake vortices' is probably too strong and would need to be moderated since this simpler model is not capturing all the dynamics. I will comment more on this later.*
   I will add the simplifications that have to be made. The BH model is also only a

tool to describe the vortex geometry in a 2D cut. This point might need additional stressing.

Introduction

2. *I would think that bringing the context of Germany appears too specific, since the wind energy sector is growing in other countries.*
This is indeed true. The introduction has been 'internationalised'.

3. *p1 l21: "In research..." this sentence and the following two are hard to read and could be reformulated*
The abstract as well as the introduction were rewritten.

4. *p2 l8: The scaling problem of wind tunnel measurements could be mentioned here*
Thank you for the hint. It will be mentioned.

5. *p2 l10: make sure the acronym for UAS (and other acronyms) is made explicit in the introduction*
Has been implemented.

6. *p2 l15: Could you mention the arguments for the offshore comparison. It probably relies on arguments on the boundary layer when the flow comes from the shore, but wave exciting the turbines and the surface roughness may be different.*
Yes, there is definitely an influence through the surface roughness compared to the sea. It can also be seen in Fig. 5 that the horizontal wind in a certain altitude experiences much more fluctuations. So there is an internal boundary (surface) layer that shows increased turbulence with a more stable marine layer on top. This was only mentioned since for the briefly presented HeliOW project it would be desirable to fly and measure behind an actual off-shore wind turbine. But for

several practical reasons this was simply not feasible.

The influence of turbulence is also a good point for future measurements to get a large amount of vortex measurements to do some statistics.

7. *p2 l18: "The project aims for safe helicopter flight paths in off-shore wind energy parks" needs reformulation*
Will be changed to "The goal of the project is also to determine save helicopter flight paths in off-shore wind energy parks".

8. *p2 l19, l21 \*University of\* Munich, University \*of\* Stuttgart.*
Has been changed accordingly.

9. *p2 l23: the wind turbine also generates strong coherent vorticies, can these be considered turbulence?*
Coherent (in this case artificial turbulent) structures are not turbulence covered by Kolmogorov 1941. The coherent vortices might be considered as a large scale anisotrop turbulence. The surrounding atmosphere in the best case scenario can be considered as isotrop. In the far wake these coherent structures decay and add to the isotrop turbulence.

10. *p2 l30: could you highlight more the difference between the study from Subramanian and yours?*
Will do. Subramanian et al.detected tip vortices by a pressure signal along a longitudinal flight path. The vortices where also not quantified (vortex strength, core radius). I will add some more details to the manuscript.

Section 2

11. *p3 l7: Wind speed and directions could have changed during this 15min period, do you have access to measurements to support this assumption?*

Unfortunately, we didn't have access to highly resolved SCADA data for the measurement time which could have given us precise information about wind speed and direction changes. Moreover, we were flying about 27 m behind a 114 m diameter WEC. If the wind direction would have changed significantly we might have crashed into the rotor blades. For a single vortex measurement it is also not important, if the wind direction varies a bit. The wind velocity component rotation later in the manuscript was done for the single measurement. The mean wind direction is only important to set up the flight trajectory for the UAS.

Section 3

12. *p5 l11: You could mention that the ring vortex is an approximation of the wake vorticity at high tip-speed ratio.*
    Thank you for the input. I will mention that.

13. *p5 l14: This line can be reformulated to mention that this result is true under the vortex ring assumption.*
    The line has been modified.

14. *p6 Figure 4: "recreation" may not be the correct word in the caption, maybe "model", or "reproduction" would be more accurate?*
    Done.

15. *Figure 7: Instead of using north/east for the axis wouldn't it be easier in that case to use an orientation in the frame of the turbine, with y pointing upstream against the main wind direction? You could then remove the sentence at the end of page 6.*
    This was actual the intent when planing the flight path at the field campaign. The wind was slowly turning. So Figure 7 shows the actual flight path (direction) of the UAS. So y pointing north is almost into the main wind direction. But at the

time of the measurement the wind direction was already a bit off. Also the post-processing software of the UAS data always writes the wind velocities according to the meteorological standards $u$ pointing east, $v$ pointing north and $w$ pointing upwards. So the data in the end always has to be rotated into the main wind direction. And this is necessary to later make $\overline{u'} = 0$ in the wake.

16. *p7 line1-12: The potential flow assumption probably appears too early in this paragraph and the paragraph could be reformulated. The definition of circulation as function of the vorticity is independent of this assumption. Equation (3) only uses an axi-symmetric assumption. It is yet true that the circulation of a vortex makes more sense in inviscid flows where the vorticity is condensed to confined singular regions.*
I will look into a possible reformulation and implementation of the introduction of potential flow.

17. *p10 l1-4: This paragraph needs should be reformulated, the language improved.*
The paragraph has been rewritten.

18. *p10 l9-15 and p11 l1-8: While reading the text I was confused since figure 9b didn't appear to be mentioned. The explanation could be improved by clearly explaining both figures and both scenario, before mentioning figure 10. Alternatively, you could tell the reader to focus only on figure 9a for now and figure 9b will be explained later. Also, equations 8-10 could be introduced first before drawing the conclusions that there is no unique solution. Further, the way the equations are introduced can be improved by telling to the reader what is coming, e.g. "The velocities at point 1 and 2 are...". Right now, they appear in the text announced.*
Figure 9b is mentioned on page 11 l7. I agree that the two different cases of passing a vortex in the UAS measurements needs additional addressing to not lose the reader.

19. *p11 l9: "This double peak can be lead back to passing the maximum tangential*

*velocity at $r = r_c$ at position 1 and 2". The sentence may need reformulation*
Changed to: This double peak is caused by passing the maximum tangential velocity at $r = r_c$ at position 1 and 2".

20. *p11 l12-15: Similar to the previous remark, the equations can be introduced before drawing the conclusions.*
I think this comes down to personal preference. I would like to explain what's happening when passing the vortex at $\Delta y < r_c$ and follow up with what that means for the previously introduces equations.

21. *p11: "As shown above the presence and identification of a vortex (or a pair of vortices) is measurable". The language needs to be reformulated (one cannot measure an identification). Also, the previous section seemed to show that in some cases the determination was not possible, which would imply that the identification is not always possible.*
Fair point. "As shown above the presence is measurable and a later identification of a vortex (or a pair of vortices) possible when the $\Delta y < r_c$ is met."

Section 4

22. *p13 l8: It would help the reader to provide some information about the measurement campaigns (how many samples were selected, what were the mean conditions), and some introduction about the samples you selected (and why you selected them) to present in the paper.*
This issue will be addressed with the additional subsection about data availability.

23. *p14 l26: It is not clear why you mention the skewed vortex at this stage. You may need to guide the reader. Also, the definition of the skewed vortex on figure 15 appears unclear. From the figure it seems that the vortex is simply rotated. I'm not sure this qualifies as a skewed vortex. Also the 2D cut appears to have some*

*3D aspect to it, which can be confusing. Introducing a coordinate system on the 3D vortex on the 2D cut can help the understanding.*

Maybe 'canted' or 'oblique' vortex is the more precise expression. The additional 3D features are meant to help the reader. That is why the blue is more transparent. It could be removed, if it confuses more than it helps?!

I have to introduce the oblique vortex hose, since the real (measurement) world is different from the analytical approximation (ring vortex). I will try to make that more clear.

24. *p14 l30: I am wondering if "analytical reconstruction" is the proper term and how this "reconstruction" is different from the previous section. The parameters you extracted from the measurements were fitted to an analytical model. The "reconstructed" vortex is this analytically modelled vortex, and it is intrinsically part of the results you presented in table 2. If I understand this correctly, it could make sense to have the modelled vortex directly on the figures 13-15, as "fitted vortices".*

In Section 3 I want to present the theory/method how to "reconstruct" the vortex from GPS and pressure data. In section 4 the individual fit is calculated. One might consider to just remove this subsection, e.g. Fuertes 2019 never introduces any vortex model of the WEC and just "fitted" the data.

As for the second point sadly the GPS standard deviation is in the range of the vortex core radius. So the double peak is not visible and overall the accuracy is not up to the in-situ, in-line pressure measurement. Thus, to keep an step-by-step approach and to use the analytical model as a proof of concept I would like to keep it separate from the measurement.

25. *p15 l9: What is meant by "artificially induced drop" of velocity? Does this refer to drop of velocity in the turbine wake? If this is so, the drop in velocity should be a function of the thrust coefficient at the rotor, and a value of 65% may not be comparable to other measurements unless they are at the same operating*

*conditions.*

You are correct, the statement has been changed. The actual deficit in percent or m/s is only important for the correction of the modelled $v$ component.

Section 5

26. *p18 eq20: You may have to introduce all symbols closer to the equation, even if these are obvious.*
    I will try to do so.

27. *p19 l1-4: It appears surprising that the authors do not have more information about the turbine (thrust curve, pitch curve). Earlier in the text, it was mentioned that a model of the turbine was done. These quantities can then be obtained from a Blade Element Momentum code.*
    *The argument here may simply be that most turbine have a pitch angle around +/- 1 degree below rated, and in the absence of data, you picked 0 degree. It is also not clear where the pitch angle enters in the equation. Most likely an argument of the thrust coefficient, but usually you'll have a thrust coefficient vs wind speed curve available for that turbine.*
    The engineering sector of wind energy converters is a highly competitive market. Although the University of Stuttgart has a generic recreation of the wind turbine, it is reserved for wake computations. The resulting aerodynamic properties of the turbine are however strongly confidential. That's why those properties, namely here the thrust coefficient, had to be estimated. As stated on p. 19 l1, we approximated the thrust coefficient vs. wind speed with an NREL 5 MW turbine. Since the E112 and NREL 5MW wind turbines both have similar rotor diameters and produce a similar amount of electricity, the approximation was judged reasonable.

28. *p19 l10-11: These sentences need to be moderated. First, it appears that the*

*study was only done on one vortex at a given operating conditions and a more quantitative analysis would be required. Second, it appears wrong to state that the wake of a turbine is described by two vortices. You could clarify your discussions based on the following considerations. The fact that two vortices are crossed on the trajectory of the drone is due to the likelihood of crossing the tip-vortices from different blades. This likelihood increases as the number of blades or the tip-speed ratio increases. When it is such, the wake vorticity surface can be approximated to a vortex cylinder, in which case any trajectory of the drone will indeed cross the tip-vorticity surface twice. This cylindrical surface does not resemble two vortices spinning in opposite direction and the wake dynamics cannot be described by assuming that it consists of two vortices. What the author probably means is that the velocity field across a tip-vortex (or a tip-vorticity surface) resembles the one of a regularized point vortex. This analogy (which is natural given the different analytical vortex wake models of wind turbines) cannot be used to "describe" the wake, but it can be used to "estimate" some of the wake properties, that is, the tip-vortex core radius and circulation.*

I agree. This paragraph needs to be more precise. I will implement your suggestions.

Section 6

29. *p19 l21: The identification of one vortex strength does not appear to be enough to draw a conclusion, or the conclusion needs to be moderated. Also, it may not be necessary to attribute this equation to Sorensen et al. and instead it can be mentioned where this formula comes from: circulation for a rotor of constant thrust coefficient (This should also be mentioned earlier in the text p18 l1-6).*

The conclusions will be readdressed and the statement moderated. Also I want to repeat at this point (and later in the new version of the conclusions) that the fact that we hit two (!) consecutive vortices in one flight leg at the crucial criterion ($\Delta y < r_\mathrm{c}$) must also be considered lucky. That both measurements, after consideration of the non-ideal conditions (canted vortex hose) and therefore the rotation of the horizontal measurements into the vortex plane is giving two circulations strengths that match the theoretical calculation, needs an appropriate (strong) statement.

---

## Author Response (AR1)

**Author's response to Referee #1**

May 21, 2019

Thank you for the detailed review of the manuscript. In the following I will comment on each point. The referee's comments will be repeated in blue italic before the answer.

**General comments**

*The manuscript by Mauz et al. describes in-situ measurements of wind turbine tip vortices with a UAS. From these measurements, circulation of the vortices is calculated using the Burnham-Hallock wake vortex model. These measurements are unique and I do not know of any other study in which a UAS operated in such close proximity to an operating wind turbine and even in its wake. The authors can convincingly show that wake vortices can be measured with the UAS. The presentation of the methods of analysis and the results however needs some significant improvement:*

- *In the introduction and throughout the text I miss more thorough references to the state-of-the-art. For example, other methods to measure full-scale wind turbine wakes with remote sensing are not mentioned at all, but are carried out all the time and in multiple ways. What can a UAS do that is not possible with remote sensing?*
  Remote sensing methods (e.g. LIDAR) can not resolve turbulence in such a small scale as a UAS is capable of. LIDARs usually cycle cone measurements that resemble averages over a huge volume compared to a UAS line measurement. However, the method of operation allows for long term measurements whereas UAS excels at in-situ small scale measurements. Appropriate literature will be added. The lack of references is also contributed to the lack of publications of remote sensing experiments that try to resolve small scale turbulence in wakes. The wakes itself can be measured by LIDAR but focus of this manuscript is the identification of tip vortices that only have been verified qualitatively by Subramanian et al.

- *The structure of the text is sometimes confusing and much information is given in the results and discussion and outlook section that should have been introduced before. I provide details in the specific comments.*
  Thank you for the feedback. We will try to improve the structure of the text. Also by implementing the specified suggestions.

- *A major problem of the manuscript is that all the analysis is only done for a single sample of a wake vortex pair. If possible at all I would strongly urge the authors to investigate if they can maybe use some other flights. Maybe even flight levels above or below hub height could still be valuable.*
  Additional vortex measurements are available. In those flight legs usually the vortex pairs are visible. However, an evaluation of the core radius and vortex strength is mainly not possible due to the distance of the UAS being larger than $r_\mathrm{c}$. From all the available data only one flight leg showed two vortices were the evaluation method presented in the manuscript was feasible.
  The aim of this manuscript is to establish an evaluation method for MASC-3 to later examine future wake measurements and then be able to establish a statistic for near wake vortex behaviour (e.g. turbulence and stratification dependent).

[Figure]

Figure 1: Wind angles and true air speed (TAS) of the 5-hole probe for the evaluated vortex measurements. Grey dashed line shows the calibration limit of −20 to 20 degrees. Overstepped angles are interpolated.

- *It is known that the estimation of the UAV attitude is a major source of error for the wind calculation. It is also known that navigation systems are less precise in dynamic flight manoeuvrers. I would therefore at least expect that attitude angles as well as airspeed and flow angles during the flight through the vortex are shown. The authors raise the issues themselves in the discussion, but I think it is necessary to include a proper analysis of this in the manuscript, including an estimation of uncertainty of the wind vector and thus the circulation.*
  The attitude angles of the UAS will be shown in an extra evaluation (e.g. subsection) also including an error estimation of the wind velocity and the measured circulation. See also Fig. 1 and Fig. 2 for a firs look.

- *I recommend some copy-editing to be done on English grammar and expressions.*
  We will look into that.

**Specific comments**

**Abstract**

*The abstract should be rewritten with more statements about the hypotheses that were investigated in the study. A description of the results / conclusion is missing.*
*p.1,ll.1ff: wind converter should be either wind turbine or wind energy converter. Preferable: "rotor blade of a wind turbine". The whole first sentence is very hard to read and grammatically wrong. Starting with the relevance for numerical models in the abstract is misleading, because numerical models are not the topic of the paper.*
*p.1,l.6: what is the difference between atmospheric and meteorological quantities?*
The abstract will be rewritten. All other annotations concerning the abstract have been implemented.

[Figure]

Figure 2: Attitude angles of the UAS for the vortex measurement flight. The entry and exit of the wake can be seen. The UAS remains on a stable flight path through the wake.

**Introduction**

*p.2,ll.9-13: this paragraph should go into the description of the aircraft (Sect. 2.1)*
*p.2,ll.14-21: this paragraph could go in the experiment/site-description in Section 2*
*p.2,ll.22-31: here, a lot more literature should be referenced. Wake vortices are a major field of research and the state-of-the-art has not been evaluated at all.*
All annotations have been addressed. Additional references and literature has been added.

**Section 2**

*I miss a detailed description of the atmospheric conditions during the flight. Stratification, turbulence in the inflow, etc. which are important for the wake development are not given but should be available from the data.*
The day the tip vortex measurement flights took place, only measurements at three different heights were made. The flight strategy was aiming on capturing tip vortices at hub height, and at the top and bottom of the wake. Since the distance to the nacelle is 0.25D or about 27 m the stratification of the atmosphere should not have a significant influence on the wake development. In an additional section a description of the atmosphere is now available, including turbulence.

*A list of available flights and an explanation why only a single flight leg is analysed is missing.*
A total of five flight legs at 0.25D are present. From these measurements only in one leg the criterion of $\Delta y < r_c$ was met. Also a new subsection has been added 'Available data' where the data availability and the atmospheric condition are mentioned.

*p.3,ll.3-8: Here, description of the aircraft is mixed with operational procedures. I feel like this should be separated.*
Content has been separated.

*p.3,l.11: Since RTK GPS is mentioned here and is not a standard feature of UAS in atmospheric research, some additional information would be appreciated: what kind of receiver is used (L1 or L1/L2 phase); is a local base station or RTCM-services used for correction data / what is the baseline? What*

[Figure]

Figure 3: Location of the E-112 WEC in the north-west of Germany near the North Sea coast. MASC flight tracks in front (blue) and in the wake (orange) of the E-112 with northern main wind direction (5 degree north that day). On the Google Earth image the WEC is oriented toward south-easterly wind direction. Map created with mapchart.net

*is the advantage of the very accurate flight path in atmospheric measurements?*
Sorry, we made a mistake. RTK was not used during the flight and will not be mentioned in the manuscript any more.

*Fig.3: I think a more schematic background (not Google Earth) with a better scale and legend would help*
Google Earth map/image was replaced with a schematic map of Germany (cf. Fig. 3).

**Section 3**

*Fig.6: It is unclear how TKE has been calculated. How large is the averaging window?*
The TKE calculation serves a qualitative purpose. Therefore an averaging of 1 s (100 data points) has been used. The integral length scale was not calculated.

*Fig.7: Nice figure, but watch out for which lines cross in front of or behind other lines to get the 3D visualization right. I think the red rectangle encloses the blue vortex, right? And "distance" should probably get a variable name or could even be left out.*
First of all, thank you for the compliments. The line issue has been addressed. They should now support the viewer's perspective. 'Distance' has been removed. An updated figure will be found in the new manuscript (cf. Fig. 4).

*p.9,l.5: Except that the vortices along the horizontal axis are not generated at the same time for the WEC.*
In the specific line it is talked about the x-axis: Along the $x$ axis, which is also the flight path of the UAS (pointing east with the main wind direction approximately 10° north), the vortices indeed show a little temporal delay. The first encountered vortex travelled a bit farther than the second vortex. This is now mentioned and could also explain the smaller core radius of the second vortex. We would like to

[Figure]

Figure 4: Simplified sketch of a vortex pair passed by the UAV to the right. In reality it would rather have a helical pattern than a ring shape. Velocities and axis according to meteorological standards, therefore axis and orientation according to the in-situ conditions. $y$ axis pointing north, $x$ axis pointing east. At hub height the $w$ component (along $z$ axis) vanishes. The red rectangle illustrates a top view of a tip vortex with distance $\Delta y$ to the UAS.

argue for simplicity reasons that the flight path was perpendicular to the wake and therefore no huge 'ageing' differences occur.

**Section 4**

*p.14,l.1: What is the vortex coordinate system and which angles are used for the rotation? This has not been introduced before.*
Thank you for this comment. 'Vortex coordinate system' might really be the wrong expression here. What we were trying to say is that the 'horizontal wind data' have been rotated into the vortex rotational plane. So after this rotation the horizontal wind plane is parallel to the rotational plane of the tip vortex. The rotation was accomplished by rotating the $x$ and $y$ axis ($u$ and $v$ wind component respectively). This information will be added in the new version of the manuscript.

*p.14, ll.26ff: this needs to be introduced and explained in more details in the methods section. Why can this correction not be done for other flight levels to increase the number of samples of tip vortices?*
In principle it is possible to look at different altitudes. The flight strategy was to concentrate on measurements at hub height, since the probability to hit a tip vortex is the highest at this level. Also the introduced simplification of the vortex only rotating in two dimensions is mainly true at hub height. The rotation into the vortex rotational plane only makes sense, when the vortex has passed with the necessary criterion ($\Delta y < r_c$). Otherwise an evaluation of $\Gamma$ and $r_c$ is not possible.

*Fig.16: I think this figure is not necessary.*
This figure has been removed.

*p.15, l.5: "rotated slightly" $\rightarrow$ what does that mean?*
Done. The wind is rotated into the wake direction.

*p.15, l.10ff: It is said that the wind speed deficit plays an important role for the tip vortex, but this is not discussed any further. Is BH even an appropriate model under these circumstances?*

The short answer is 'yes'. The $u$ component of the model is not affected by the deficit, especially since the data has been rotated into the wake direction. Also the peak position (on the $x$ axis) for the $v$ component in the BH model is also not affected. Only the slope and magnitude.

*Fig.18: not sure if this figure is necessary*
Will be dropped from the manuscript. It was simply thought to be a visualisation of the applied correction.

**Section 5**

*p.19,l.7f: These are too strong statements for an experiment with a single sample*

*p.19,l.15ff: The issues that are raised here are not insignificant and call for some more analysis and quantification.*
Attitude angles and true air speed variations will be analysed and the results addressed accordingly in the new manuscript.

**Section 6**

*p.19,l.21: What is the equation by Sorensen et al. (2014) that is meant here?*
Done. Eq. 20. Reference has been added.

*p.19,l.23: "aggravates an evaluation" - what do you mean by that?*
The evaluation is done mainly graphically. The second tip vortex is embedded in a relatively high level of turbulence (wake deficit, shear, etc.). The second tip vortex does not show a clear border to the undisturbed atmosphere as tip vortex 1 does. The reference velocity levels for the evaluation are therefore harder to extract from the measurements.

*p.19,ll.27ff: The information about the 5-hole probe calibration range and why other flights could not be used should be given in Section 2 already. In section 2, it was said that the UAV operated with RTK GPS which is contradicted here.* The subsection 'Data availability' has been added in Section 2 where we explain briefly why no other vortex examples are available. The RTK GPS mentioning will be stripped. We will link to a recent MASC-3 paper by Rautenberg et al. 2019.

**Technical corrections**

*p.1,l.1ff: wind converter should be either wind turbine or wind energy converter. Preferable: "rotor blade of a wind turbine". The whole first sentence is very hard to read and grammatically wrong. The second sentence raises problems of numerical simulations that are not the topic of the paper.*
*p.1,l.6: what is the difference between atmospheric and meteorological quantities?*
*p.1,l.8: u,v,w, italic please*
*p.2,l.3: underestimate*
*p.2,l.4: "Another way" or "Another method"*
*p.8,l.7: "several analytical models are available"!?*
*Fig.8: x-axis should be labelled r*
*p.11,l.18: L is not proportional to the velocity ratio → The velocity ratio is a function of L.*
*p.14,l.10f: "In Fig.14 shown as a solid purple line." What is?*
*p.14,l.21: "clear and sharp jump" - strange expression*
*p.14,l.22: Fig. 16 appears before Fig. 15 in the text.*
*Fig.17: What is u and what is v should be mentioned in the caption. the same line style should be used for the same analysis method, i.e. dashed line for simple BH, and dotted for corrected version for example.*
All corrections have been adopted and implemented in the new manuscript.

*p.1,l.18: "their individual capacity and diversity" (what do you mean by diversity?)*
Here we wanted to point out that there are different WEC designs for different terrain (complex vs. homogeneous) with all their challenges (high wind speeds, high turbulence and increased stress on blade structures etc.) The paragraph has been rewritten.

**Author's response to Referee #2**

May 21, 2019

Thank you for the detailed review of the manuscript. In the following I will comment on each point. The referee's comments will be repeated in blue italic before the answer.

**General comments**

*I agree with the comments from RC1, so I will focus on some of the other things I noted. I have written down some general comments below, followed by a list of specific comments. As it is, I found the article hard to follow, which is unfortunate since the results are interesting. The authors should focus on guiding the reader through their thoughts and results, in a clear and concise way, reducing the length of the paper. This will require a major revision of the paper. I also believe the authors should spend an important amount on time correcting the language of the article. I've highlighted a couple of paragraph and sentences that*

- *I also believe the authors should spend an important amount on time correcting the language of the article. I've highlighted a couple of paragraph and sentences that could be improved up to page 11, after which I stopped mentioning these. The authors should still work on the text after page 11.*
  We will look into that.

- *Regarding the title, I would probably not support using the word 'first' in the title since it brings a competitive touch to it that is unnecessary in my opinion. Also in light of the following publication, it may unfortunately not be justified to claim this 'first' attribute: F. Carbajo Fuertes et al - 2019 - 'Multirotor UAV based platform for the measurement of atmospheric turbulence: validation and signature detect ion of tip vortices of wind turbine blades.'. The author may also consider the studies from, Kocer et al. 2011 and Reuder and Jonassen 2012 cited in the above reference. (Please note that I'm not an author of any of these papers). Yet, I leave this up to the authors and the editor to decide whether to change the title.*
  Thank you for your valuable input there. The paper by Carbajo Fuertes was put online on March 28 2019. So there was no possibility to the authors to have known from these results.
  Following your suggestion I had a look at Fuertes 2019. The results look good. It is nice to see that two different approaches and measurement systems come to a similar result and output. Fuertes also derives a circulation of 50 $m^2/s$ from his measurements. Given that he measured at a different wind turbine, the circulation strength is of the same order of magnitude. However, I could not find any information about the vortex fit he is using for his measurements as well as a detailed information about the circulation calculation.

- *I would personally prefer the equations to be closer to the text. As it is now, the equations are usually floating at the end of the paragraph which can make the discussions hard to follow.*
  I will consider this point to be mainly to be an editorial problem. I will see, to put equations closer to the text. However, in the later two-column layout everything will change again. So I might need input from the editor at this point.

- *It appears that the method presented can be attributed to Fischenberg, and the author may need to be clearer when highlighting if something is new or unique in their approach compared to what was already published (apart from the measurement campaign). The model using a regularized vortex cannot really be seen as a new contribution or method. The experimental data though are of high value.*

  It is correct that the method and model is not new. But as we state that the model has its origin in aviation, it is new to be used to describe wind turbine wake vortices properties. We added a paragraph to highlight this new approach in WEC wake studying.

  Also the described method illustrates how to obtain results from in-situ line measurements of a UAS, including systematic measurement uncertainties like skewed/canted vortex hoses etc.

- *The amount of measurements appear unclear, and some statistical analysis and information about the ensemble of results available could be valuable. Reading the article, it seems that only one vortex was analysed. Further data with different distances downstream should be incorporated since according to section 2 more data was acquired. Statistical tools should also be used to mention the uncertainty on the fitted parameters and to quantify the error between the model and the measurements.*

  A new subsection '2.3 Available data' has been added. We also state why only one pair of vortices could be used for an evaluation. A simple vortex measurement in the wake is not sufficient, also the criterion $\Delta y < r_c$ must be met which is a rare condition. We comment on this in the 'specific comment' section below in more detail.

- *The figures are usually clear. The authors could yet reduce the number of figures, particularly in the first 10 pages, or by combining the measurements with the fitted model in figures 10-17.*

  We will take out some figures from the manuscript. But also some new ones will be added. Mainly to do the uncertainty evaluation (wind angles, true air speed, etc.). We will also try if some figures could be combined. But that might add to the readers confusion.

- *I hope my numerous comments will not discourage the authors, and I strongly encourage them to further work on this paper. As I mentioned earlier, the article has some great potential, it just needs some additional work to reach the point of publication.*

  Highly appreciated emphasis. Thank you!

**Specific comments**

**Abstract**

*The statement in the abstract 'the BH model can be used to describe wake vortices' is probably too strong and would need to be moderated since this simpler model is not capturing all the dynamics. I will comment more on this later.*

We will add the simplifications that have to be made. The BH model is also only a tool to describe the vortex geometry in a 2D cut. This point might need additional stressing.

**Introduction**

*I would think that bringing the context of Germany appears too specific, since the wind energy sector is growing in other countries.*

This is indeed true. The introduction has been 'internationalised'.

*p1 l21: "In research..." this sentence and the following two are hard to read and could be reformulated*

The abstract as well as the introduction were rewritten.

*p2 l8: The scaling problem of wind tunnel measurements could be mentioned here*
Thank you for the hint. It will be mentioned.

*p2 l10: make sure the acronym for UAS (and other acronyms) is made explicit in the introduction*
Has been implemented.

*p2 l15: Could you mention the arguments for the offshore comparison. It probably relies on arguments on the boundary layer when the flow comes from the shore, but wave exciting the turbines and the surface roughness may be different.*
Yes, there is definitely an influence through the surface roughness compared to the sea. It can also be seen in Fig. 5 that the horizontal wind in a certain altitude experiences much more fluctuations. So there is an internal boundary (surface) layer that shows increased turbulence with a more stable marine layer on top.
This was only mentioned since for the briefly presented HeliOW project it would be desirable to fly and measure behind an actual off-shore wind turbine. But for several practical reasons this was simply not feasible.
The influence of turbulence is also a good point for future measurements to get a large amount of vortex measurements to do some statistics.

*p2 l18: "The project aims for safe helicopter flight paths in off-shore wind energy parks" needs reformulation*
Will be changed to "The goal of the project is also to determine safe helicopter flight paths in off-shore wind energy parks".

*p2 l19, l21 \*University of\* Munich, University \*of\* Stuttgart.*
Has been changed accordingly.

*p2 l23: the wind turbine also generates strong coherent vorticies, can these be considered turbulence?*
Coherent (in this case artificial turbulent) structures are not turbulence covered by Kolmogorov 1941. The coherent vortices might be considered as a large scale anisotrop turbulence. The surrounding atmosphere in the best case scenario can be considered as isotrop. In the far wake these coherent structures decay and add to the isotrop turbulence.

*p2 l30: could you highlight more the difference between the study from Subramanian and yours?*
Subramanian et al. detected tip vortices by a pressure signal along a longitudinal flight path. The vortices were also not quantified (vortex strength, core radius). We will add some more details to the manuscript and highlight the main differences.

**Section 2**

*p3 l7: Wind speed and directions could have changed during this 15min period, do you have access to measurements to support this assumption?*
Unfortunately, we didn't have access to highly resolved SCADA data for the measurement time which could have given us precise information about wind speed and direction changes. Moreover, we were flying about 27 m behind a 114 m diameter WEC. If the wind direction would have changed significantly we might have crashed into the rotor blades. For a single vortex measurement it is also not important, if the wind direction varies a bit. The wind velocity component rotation later in the manuscript was done for the single measurement. The mean wind direction is only important to set up the flight trajectory for the UAS.

**Section 3**

*p5 l11: You could mention that the ring vortex is an approximation of the wake vorticity at high tip-speed ratio.*
Thank you for the input. We will mention that.

*p5 l14: This line can be reformulated to mention that this result is true under the vortex ring assumption.*
The line has been modified.

Done.

*Figure 7: Instead of using north/east for the axis wouldn't it be easier in that case to use an orientation in the frame of the turbine, with y pointing upstream against the main wind direction? You could then remove the sentence at the end of page 6.*

This was actual the intent when planing the flight path at the field campaign. The wind was slowly turning. So Figure 7 shows the actual flight path (direction) of the UAS. So y pointing north is almost into the main wind direction. But at the time of the measurement the wind direction was already a bit off. Also the post-processing software of the UAS data always writes the wind velocities according to the meteorological standards $u$ pointing east, $v$ pointing north and $w$ pointing upwards. So the data in the end always has to be rotated into the main wind direction. And this is necessary to later make $\overline{u'} = 0$ in the wake.

*p7 line1-12: The potential flow assumption probably appears too early in this paragraph and the paragraph could be reformulated. The definition of circulation as function of the vorticity is independent of this assumption. Equation (3) only uses an axi-symmetric assumption. It is yet true that the circulation of a vortex makes more sense in inviscid flows where the vorticity is condensed to confined singular regions.*

We will look into a possible reformulation and implementation of the introduction of potential flow.

*p10 l1-4: This paragraph needs should be reformulated, the language improved.*

The paragraph has been rewritten.

*p10 l9-15 and p11 l1-8: While reading the text I was confused since figure 9b didn't appear to be mentioned. The explanation could be improved by clearly explaining both figures and both scenario, before mentioning figure 10. Alternatively, you could tell the reader to focus only on figure 9a for now and figure 9b will be explained later. Also, equations 8-10 could be introduced first before drawing the conclusions that there is no unique solution. Further, the way the equations are introduced can be improved by telling to the reader what is coming, e.g. "The velocities at point 1 and 2 are...". Right now, they appear in the text announced.*

Figure 9b is mentioned on page 11 l7. I agree that the two different cases of passing a vortex in the UAS measurements needs additional addressing to not lose the reader. The paragraphs have been modified.

*p11 l9: "This double peak can be lead back to passing the maximum tangential velocity at $r = r_c$ at position 1 and 2". The sentence may need reformulation*

Changed to: This double peak is caused by passing the maximum tangential velocity at $r = r_c$ at position 1 and 2".

*p11 l12-15: Similar to the previous remark, the equations can be introduced before drawing the conclusions.*

We think this comes down to personal preference. We would like to explain what's happening when passing the vortex at $\Delta y < r_c$ and follow up with what that means for the previously introduces equations.

*p11: "As shown above the presence and identification of a vortex (or a pair of vortices) is measurable". The language needs to be reformulated (one cannot measure an identification). Also, the previous section seemed to show that in some cases the determination was not possible, which would imply that the identification is not always possible.*

Fair point. "As shown above the presence is measurable and a later identification of a vortex (or a pair of vortices) possible when the $\Delta y < r_c$ is met."

**Section 4**

*p13 l8: It would help the reader to provide some information about the measurement campaigns (how many samples were selected, what were the mean conditions), and some introduction about the samples you selected (and why you selected them) to present in the paper.*

This issue will be addressed with the additional subsection about data availability.

Maybe 'canted' or 'oblique' vortex is the more precise expression. The additional 3D features are meant to help the reader. That is why the blue is more transparent. It could be removed, if it confuses more than it helps?!

We have to introduce the oblique vortex hose, since the real (measurement) world is different from the analytical approximation (ring vortex). We will try to make that more clear.

In Section 3 we want to present the theory/method how to "reconstruct" the vortex from GPS and pressure data. In section 4 the individual fit is calculated. One might consider to just remove this subsection, e.g. Fuertes 2019 never introduces any vortex model of the WEC and just "fitted" the data. As for the second point, in fact the GPS positioning standard deviation is in the range of the vortex core radius. So the double peak is not visible and overall the accuracy is not up to the in-situ, in-line pressure measurement. Thus, to keep an step-by-step approach and to use the analytical model, as a proof of concept, we prefer to keep it separate from the measurement.

You are correct, the statement has been changed. The actual deficit in percent or m/s is only important for the correction of the modelled $v$ component.

**Section 5**

We will try to do so.

The engineering sector of wind energy converters is a highly competitive market. Although the University of Stuttgart has a generic recreation of the wind turbine, it is reserved for wake computations. The resulting aerodynamic properties of the turbine are however strongly confidential. That's why those properties, namely here the thrust coefficient, had to be estimated. As stated on p. 19 l1, we approximated the thrust coefficient vs. wind speed with an NREL 5 MW turbine. Since the E112 and NREL 5MW wind turbines both have similar rotor diameters and produce a similar amount of electricity, the approximation was judged reasonable.

*vorticity surface can be approximated to a vortex cylinder, in which case any trajectory of the drone will indeed cross the tip-vorticity surface twice. This cylindrical surface does not resemble two vortices spinning in opposite direction and the wake dynamics cannot be described by assuming that it consists of two vortices. What the author probably means is that the velocity field across a tip-vortex (or a tip-vorticity surface) resembles the one of a regularized point vortex. This analogy (which is natural given the different analytical vortex wake models of wind turbines) cannot be used to "describe" the wake, but it can be used to "estimate" some of the wake properties, that is, the tip-vortex core radius and circulation.*
We agree. This paragraph needs to be more precise. We will implement your suggestions.

**Section 6**

*p19 l21: The identification of one vortex strength does not appear to be enough to draw a conclusion, or the conclusion needs to be moderated. Also, it may not be necessary to attribute this equation to Sorensen et al. and instead it can be mentioned where this formula comes from: circulation for a rotor of constant thrust coefficient (This should also be mentioned earlier in the text p18 l1-6).*
The conclusions will be readdressed and the statement moderated. Also we want to repeat at this point (and later in the new version of the conclusions) that the fact that we hit two (!) consecutive vortices in one flight leg at the crucial criterion ($\Delta y < r_c$) might also be considered lucky. That both measurements, after consideration of the non-ideal conditions (canted vortex hose) and therefore the rotation of the horizontal measurements into the vortex plane is giving two circulations strengths that match the theoretical calculation, needs an appropriate (strong) statement.

[revised manuscript text omitted]
_\text{t,2} = V_\text{t,1} = V_\text{t,max} = \frac{\Gamma}{2\pi} \frac{r_\text{c}}{r_\text{c}^2 + r_\text{c}^2} = \frac{\Gamma}{4\pi r_\text{c}}\tag{11}$$

$$V_\text{t,\Delta y} = \frac{\Gamma}{2\pi} \frac{\Delta y}{r_\text{c}^2 + \Delta y^2}\tag{12}$$

30 $$r_1^2 = r_2^2 = L^2 + \Delta y^2 = r_\text{c}^2 \longleftrightarrow \Delta y^2 = r_\text{c}^2 - L^2\tag{13}$$

[Figure]

**Figure 10.** Analytical solution of a UAS passing the vortex at a path crossing the centre (black solid line), passing at $r = 0.5r_c$ (red line) and at a distance double the core radius (black dashed line). The peak to peak distance is $2\,L$ (cf. Fig. 9), above illustrated for the black solid line.

With now only 3 ($\Gamma$, $r_c$ and $\Delta y$) unknown parameters it is possible to solve the equations.

Dividing Eq. 12 by Eq. 11 eliminates $\Gamma$. Inserting Eq. 13 gives:

$$\frac{V_{t,\Delta y}}{V_{t,\max}} = \frac{r_c\sqrt{r_c^2 - L^2}}{r_c^2 - \frac{L^2}{2}} \tag{14}$$

Equation 14 describes a tangential velocity ratio that is a function of $L$. Also $L$ is known to range from 0 to $r_c$. A dimensionless

5   relationship $L\,r_c^{-1}$ can be plotted and is shown in Fig. 11. By passing the vortex with $\Delta y < r_c$, and plotting the measured $V_t$ against the distance to the vortex (Fig. 10), we can determine $L, V_{t,\max}, V_t, \Delta y$. Using diagram Fig.11, we finally determine $L\,r_c^{-1}$ and thus $r_c$.

**3.3 Analytical reconstruction**

As shown above, blade-tip vortices are theoretically measurable. They can be identified by their distinct 'dent' feature when

10   the $\Delta y < r_c$ criterion is met. Basic geometry and the Burnham-Hallock model further allow for a reconstruction (analytical solution) of the individually measured vortex, which is helpful to verify the measurements and evaluation technique. With Eq. 15 and 16 the distance to each vortex core (centre), to and along the UAV flight path, can be calculated (Fischenberg, 2011). In Fig. 12 the distances of the UAS to two vortices spinning in opposite direction are shown. In the figure, vortex 1 is passed through its core and vortex 2 is passed with a slight off-set. The flight path of the UAS is indicated with a red dashed line.

15   Those distances are inserted into Eq. 17 and 18 using the relation of Eq. 4 the tangential velocity along the meteorological $x$ axis ($u$' component) and $y$ axis ($v$' component) can be calculated:

$$d_1 = \sqrt{\Delta x^2 + \Delta y^2} = \sqrt{(x - x_{\text{Vortex1}})^2 + (y - y_{\text{Vortex1}})^2} \tag{15}$$

[Figure]

**Figure 11.** Dimensionless relationship between the ratio of the minimum (dent) tangential velocity and the maximum tangential velocity versus half the peak to peak distance ($L$), in percentage of $r_c$.

[Figure]

**Figure 12.** Qualitative example of an ideal flight path (vortex 1) and a passing with a little off-set (vortex 2) of the UAS. For the field measurement the distances $d1$ and $d2$ are calculated from the UAS GPS position and the location (off-set) of the vortex in relative coordinates with WEC at $(0,0)$, indicated with a black dot. The vortex position can be derived from the extent of the tangential velocity $V_t$ measured by the UAS and the peak to peak distance, explained in the previous sections. In this example $d_{1,2} = \sqrt{\Delta x^2 + \Delta y^2}$ with $\Delta y_1 = 0$ for $d1$ and $\Delta y_2 = \text{const.} \neq 0$ for $d2$.

$$d_2 = \sqrt{\Delta x^2 + \Delta y^2} = \sqrt{(x - x_{\text{Vortex2}})^2 + (y - y_{\text{Vortex2}})^2} \tag{16}$$

While the $y$ coordinate can be derived from the measurement (using $\Delta y$ and the UAS position, s.a. chapter 4.3) the $x$ coordinate of the vortex $x_{\text{Vortex1,2}}$ is the $x$ coordinate of the flight path at the position '3', e.g. Fig. 9.

$$u' = V_{\text{t}}(d_1)\left(\frac{y - y_{\text{Vortex1}}}{d_1}\right) - V_{\text{t}}(d_2)\left(\frac{y - y_{\text{Vortex2}}}{d_2}\right) \tag{17}$$

$$v' = -V_{\text{t}}(d_1)\left(\frac{x - x_{\text{Vortex1}}}{d_1}\right) + V_{\text{t}}(d_2)\left(\frac{x - x_{\text{Vortex2}}}{d_2}\right) \tag{18}$$

**4 Results**

**4.1 Vortex measurement**

Figure 13a shows the $v_{\text{h}} = \sqrt{u^2 + v^2}$ component of the wind measurement behind the WEC at hub height. The data reveals several (near) wake specific features. This flight leg shows two measurements of a tip vortex, indicated by the arrows in Fig. 13a. In-between those two peaks the wake deficit is measurable by a significant drop of the horizontal wind velocity. Due to the near vicinity to the nacelle, the wake deficit is dominated by turbulence created by the blade root vortices.

Figure 15 shows a zoomed-in look at the measured vortices depicted in Fig. 13. Figure 15a shows the vortex measured while entering the wake (vortex 1) and 15b while leaving the wake (vortex 2). Both, (a) and (b), are the plain UAS measurements. Figure 15c and Fig. 15d show the same measurement but the UAS coordinate system is rotated into the vortex rotational plane. This ensures that the rotational energy of the vortex is entirely captured by the $u$ and $v$ component, thus becoming the velocity components of the two dimensional rotational plane of the vortex, shown in Fig. 15 as a solid purple line. Examining both vortices, the velocity distribution pattern of the UAS passing at distance $r < r_{\text{c}}$ is visible in the $v_{\text{h}}$ measurement. The horizontal wind velocity $v_{\text{
[revised manuscript text omitted]

Major changes in the manuscript:

- Abstract and Introduction rewritten
- New subsection "Available data" and Paragraph to research UAS have been rewritten
- Improved precision of certain statements all over the text. E.g. in the method section and discussion/outlook
- Several Images have been removed
- New subsection "Quality control and error estimation" has been added:
  --> error estimation from UAS attitude and flow angles
- Language and grammar improvements

**Author's response to Referee #1**

May 21, 2019

Thank you for the detailed review of the manuscript. In the following I will comment on each point. The referee's comments will be repeated in blue italic before the answer.

**General comments**

*The manuscript by Mauz et al. describes in-situ measurements of wind turbine tip vortices with a UAS. From these measurements, circulation of the vortices is calculated using the Burnham-Hallock wake vortex model. These measurements are unique and I do not know of any other study in which a UAS operated in such close proximity to an operating wind turbine and even in its wake. The authors can convincingly show that wake vortices can be measured with the UAS. The presentation of the methods of analysis and the results however needs some significant improvement:*

- *In the introduction and throughout the text I miss more thorough references to the state-of-the-art. For example, other methods to measure full-scale wind turbine wakes with remote sensing are not mentioned at all, but are carried out all the time and in multiple ways. What can a UAS do that is not possible with remote sensing?*

  Remote sensing methods (e.g. LIDAR) can not resolve turbulence in such a small scale as a UAS is capable of. LIDARs usually cycle cone measurements that resemble averages over a huge volume compared to a UAS line measurement. However, the method of operation allows for long term measurements whereas UAS excels at in-situ small scale measurements. Appropriate literature will be added. The lack of references is also contributed to the lack of publications of remote sensing experiments that try to resolve small scale turbulence in wakes. The wakes itself can be measured by LIDAR but focus of this manuscript is the identification of tip vortices that only have been verified qualitatively by Subramanian et al.

- *The structure of the text is sometimes confusing and much information is given in the results and discussion and outlook section that should have been introduced before. I provide details in the specific comments.*

  Thank you for the feedback. We will try to improve the structure of the text. Also by implementing the specified suggestions.

- *A major problem of the manuscript is that all the analysis is only done for a single sample of a wake vortex pair. If possible at all I would strongly urge the authors to investigate if they can maybe use some other flights. Maybe even flight levels above or below hub height could still be valuable.*

  Additional vortex measurements are available. In those flight legs usually the vortex pairs are visible. However, an evaluation of the core radius and vortex strength is mainly not possible due to the distance of the UAS being larger than $r_\mathrm{c}$. From all the available data only one flight leg showed two vortices were the evaluation method presented in the manuscript was feasible.

  The aim of this manuscript is to establish an evaluation method for MASC-3 to later examine future wake measurements and then be able to establish a statistic for near wake vortex behaviour (e.g. turbulence and stratification dependent).

[Figure]

Figure 1: Wind angles and true air speed (TAS) of the 5-hole probe for the evaluated vortex measurements. Grey dashed line shows the calibration limit of −20 to 20 degrees. Overstepped angles are interpolated.

- *It is known that the estimation of the UAV attitude is a major source of error for the wind calculation. It is also known that navigation systems are less precise in dynamic flight manoeuvrers. I would therefore at least expect that attitude angles as well as airspeed and flow angles during the flight through the vortex are shown. The authors raise the issues themselves in the discussion, but I think it is necessary to include a proper analysis of this in the manuscript, including an estimation of uncertainty of the wind vector and thus the circulation.*
  The attitude angles of the UAS will be shown in an extra evaluation (e.g. subsection) also including an error estimation of the wind velocity and the measured circulation. See also Fig. 1 and Fig. 2 for a firs look.

- *I recommend some copy-editing to be done on English grammar and expressions.*
  We will look into that.

**Specific comments**

**Abstract**

*The abstract should be rewritten with more statements about the hypotheses that were investigated in the study. A description of the results / conclusion is missing.*
*p.1,ll.1ff: wind converter should be either wind turbine or wind energy converter. Preferable: "rotor blade of a wind turbine". The whole first sentence is very hard to read and grammatically wrong. Starting with the relevance for numerical models in the abstract is misleading, because numerical models are not the topic of the paper.*
*p.1,l.6: what is the difference between atmospheric and meteorological quantities?*
The abstract will be rewritten. All other annotations concerning the abstract have been implemented.

[Figure]

Figure 2: Attitude angles of the UAS for the vortex measurement flight. The entry and exit of the wake can be seen. The UAS remains on a stable flight path through the wake.

**Introduction**

*p.2,ll.9-13: this paragraph should go into the description of the aircraft (Sect. 2.1)*
*p.2,ll.14-21: this paragraph could go in the experiment/site-description in Section 2*
*p.2,ll.22-31: here, a lot more literature should be referenced. Wake vortices are a major field of research and the state-of-the-art has not been evaluated at all.*
All annotations have been addressed. Additional references and literature has been added.

**Section 2**

*I miss a detailed description of the atmospheric conditions during the flight. Stratification, turbulence in the inflow, etc. which are important for the wake development are not given but should be available from the data.*
The day the tip vortex measurement flights took place, only measurements at three different heights were made. The flight strategy was aiming on capturing tip vortices at hub height, and at the top and bottom of the wake. Since the distance to the nacelle is 0.25D or about 27 m the stratification of the atmosphere should not have a significant influence on the wake development. In an additional section a description of the atmosphere is now available, including turbulence.

*A list of available flights and an explanation why only a single flight leg is analysed is missing.*
A total of five flight legs at 0.25D are present. From these measurements only in one leg the criterion of $\Delta y < r_c$ was met. Also a new subsection has been added 'Available data' where the data availability and the atmospheric condition are mentioned.

*p.3,ll.3-8: Here, description of the aircraft is mixed with operational procedures. I feel like this should be separated.*
Content has been separated.

*p.3,l.11: Since RTK GPS is mentioned here and is not a standard feature of UAS in atmospheric research, some additional information would be appreciated: what kind of receiver is used (L1 or L1/L2 phase); is a local base station or RTCM-services used for correction data / what is the baseline? What*

[Figure]

Figure 3: Location of the E-112 WEC in the north-west of Germany near the North Sea coast. MASC flight tracks in front (blue) and in the wake (orange) of the E-112 with northern main wind direction (5 degree north that day). On the Google Earth image the WEC is oriented toward south-easterly wind direction. Map created with mapchart.net

*is the advantage of the very accurate flight path in atmospheric measurements?*
Sorry, we made a mistake. RTK was not used during the flight and will not be mentioned in the manuscript any more.

*Fig.3: I think a more schematic background (not Google Earth) with a better scale and legend would help*
Google Earth map/image was replaced with a schematic map of Germany (cf. Fig. 3).

**Section 3**

*Fig.6: It is unclear how TKE has been calculated. How large is the averaging window?*
The TKE calculation serves a qualitative purpose. Therefore an averaging of 1 s (100 data points) has been used. The integral length scale was not calculated.

*Fig.7: Nice figure, but watch out for which lines cross in front of or behind other lines to get the 3D visualization right. I think the red rectangle encloses the blue vortex, right? And "distance" should probably get a variable name or could even be left out.*
First of all, thank you for the compliments. The line issue has been addressed. They should now support the viewer's perspective. 'Distance' has been removed. An updated figure will be found in the new manuscript (cf. Fig. 4).

*p.9,l.5: Except that the vortices along the horizontal axis are not generated at the same time for the WEC.*
In the specific line it is talked about the x-axis: Along the $x$ axis, which is also the flight path of the UAS (pointing east with the main wind direction approximately 10° north), the vortices indeed show a little temporal delay. The first encountered vortex travelled a bit farther than the second vortex. This is now mentioned and could also explain the smaller core radius of the second vortex. We would like to

[Figure]

Figure 4: Simplified sketch of a vortex pair passed by the UAV to the right. In reality it would rather have a helical pattern than a ring shape. Velocities and axis according to meteorological standards, therefore axis and orientation according to the in-situ conditions. $y$ axis pointing north, $x$ axis pointing east. At hub height the $w$ component (along $z$ axis) vanishes. The red rectangle illustrates a top view of a tip vortex with distance $\Delta y$ to the UAS.

argue for simplicity reasons that the flight path was perpendicular to the wake and therefore no huge 'ageing' differences occur.

**Section 4**

*p.14,l.1: What is the vortex coordinate system and which angles are used for the rotation? This has not been introduced before.*
Thank you for this comment. 'Vortex coordinate system' might really be the wrong expression here. What we were trying to say is that the 'horizontal wind data' have been rotated into the vortex rotational plane. So after this rotation the horizontal wind plane is parallel to the rotational plane of the tip vortex. The rotation was accomplished by rotating the $x$ and $y$ axis ($u$ and $v$ wind component respectively). This information will be added in the new version of the manuscript.

*p.14, ll.26ff: this needs to be introduced and explained in more details in the methods section. Why can this correction not be done for other flight levels to increase the number of samples of tip vortices?*
In principle it is possible to look at different altitudes. The flight strategy was to concentrate on measurements at hub height, since the probability to hit a tip vortex is the highest at this level. Also the introduced simplification of the vortex only rotating in two dimensions is mainly true at hub height. The rotation into the vortex rotational plane only makes sense, when the vortex has passed with the necessary criterion ($\Delta y < r_c$). Otherwise an evaluation of $\Gamma$ and $r_c$ is not possible.

*Fig.16: I think this figure is not necessary.*
This figure has been removed.

*p.15, l.5: "rotated slightly" → what does that mean?*
Done. The wind is rotated into the wake direction.

*p.15, l.10ff: It is said that the wind speed deficit plays an important role for the tip vortex, but this is not discussed any further. Is BH even an appropriate model under these circumstances?*

The short answer is 'yes'. The $u$ component of the model is not affected by the deficit, especially since the data has been rotated into the wake direction. Also the peak position (on the $x$ axis) for the $v$ component in the BH model is also not affected. Only the slope and magnitude.

*Fig.18: not sure if this figure is necessary*
Will be dropped from the manuscript. It was simply thought to be a visualisation of the applied correction.

**Section 5**

*p.19,l.7f: These are too strong statements for an experiment with a single sample*

*p.19,l.15ff: The issues that are raised here are not insignificant and call for some more analysis and quantification.*
Attitude angles and true air speed variations will be analysed and the results addressed accordingly in the new manuscript.

**Section 6**

*p.19,l.21: What is the equation by Sorensen et al. (2014) that is meant here?*
Done. Eq. 20. Reference has been added.

*p.19,l.23: "aggravates an evaluation" - what do you mean by that?*
The evaluation is done mainly graphically. The second tip vortex is embedded in a relatively high level of turbulence (wake deficit, shear, etc.). The second tip vortex does not show a clear border to the undisturbed atmosphere as tip vortex 1 does. The reference velocity levels for the evaluation are therefore harder to extract from the measurements.

*p.19,ll.27ff: The information about the 5-hole probe calibration range and why other flights could not be used should be given in Section 2 already. In section 2, it was said that the UAV operated with RTK GPS which is contradicted here.* The subsection 'Data availability' has been added in Section 2 where we explain briefly why no other vortex examples are available. The RTK GPS mentioning will be stripped. We will link to a recent MASC-3 paper by Rautenberg et al. 2019.

**Technical corrections**

*p.1,l.1ff: wind converter should be either wind turbine or wind energy converter. Preferable: "rotor blade of a wind turbine". The whole first sentence is very hard to read and grammatically wrong. The second sentence raises problems of numerical simulations that are not the topic of the paper.*
*p.1,l.6: what is the difference between atmospheric and meteorological quantities?*
*p.1,l.8: u,v,w, italic please*
*p.2,l.3: underestimate*
*p.2,l.4: "Another way" or "Another method"*
*p.8,l.7: "several analytical models are available"!?*
*Fig.8: x-axis should be labelled r*
*p.11,l.18: L is not proportional to the velocity ratio → The velocity ratio is a function of L.*
*p.14,l.10f: "In Fig.14 shown as a solid purple line." What is?*
*p.14,l.21: "clear and sharp jump" - strange expression*
*p.14,l.22: Fig. 16 appears before Fig. 15 in the text.*
*Fig.17: What is u and what is v should be mentioned in the caption. the same line style should be used for the same analysis method, i.e. dashed line for simple BH, and dotted for corrected version for example.*
All corrections have been adopted and implemented in the new manuscript.

*p.1,l.18: "their individual capacity and diversity" (what do you mean by diversity?)*
Here we wanted to point out that there are different WEC designs for different terrain (complex vs. homogeneous) with all their challenges (high wind speeds, high turbulence and increased stress on blade structures etc.) The paragraph has been rewritten.

**Author's response to Referee #2**

May 21, 2019

Thank you for the detailed review of the manuscript. In the following I will comment on each point. The referee's comments will be repeated in blue italic before the answer.

**General comments**

*I agree with the comments from RC1, so I will focus on some of the other things I noted. I have written down some general comments below, followed by a list of specific comments. As it is, I found the article hard to follow, which is unfortunate since the results are interesting. The authors should focus on guiding the reader through their thoughts and results, in a clear and concise way, reducing the length of the paper. This will require a major revision of the paper. I also believe the authors should spend an important amount on time correcting the language of the article. I've highlighted a couple of paragraph and sentences that*

- *I also believe the authors should spend an important amount on time correcting the language of the article. I've highlighted a couple of paragraph and sentences that could be improved up to page 11, after which I stopped mentioning these. The authors should still work on the text after page 11.*
  We will look into that.

- *Regarding the title, I would probably not support using the word 'first' in the title since it brings a competitive touch to it that is unnecessary in my opinion. Also in light of the following publication, it may unfortunately not be justified to claim this 'first' attribute: F. Carbajo Fuertes et al - 2019 - 'Multirotor UAV based platform for the measurement of atmospheric turbulence: validation and signature detect ion of tip vortices of wind turbine blades.'. The author may also consider the studies from, Kocer et al. 2011 and Reuder and Jonassen 2012 cited in the above reference. (Please note that I'm not an author of any of these papers). Yet, I leave this up to the authors and the editor to decide whether to change the title.*
  Thank you for your valuable input there. The paper by Carbajo Fuertes was put online on March 28 2019. So there was no possibility to the authors to have known from these results.
  Following your suggestion I had a look at Fuertes 2019. The results look good. It is nice to see that two different approaches and measurement systems come to a similar result and output. Fuertes also derives a circulation of 50 $m^2/s$ from his measurements. Given that he measured at a different wind turbine, the circulation strength is of the same order of magnitude. However, I could not find any information about the vortex fit he is using for his measurements as well as a detailed information about the circulation calculation.

- *I would personally prefer the equations to be closer to the text. As it is now, the equations are usually floating at the end of the paragraph which can make the discussions hard to follow.*
  I will consider this point to be mainly to be an editorial problem. I will see, to put equations closer to the text. However, in the later two-column layout everything will change again. So I might need input from the editor at this point.

- *It appears that the method presented can be attributed to Fischenberg, and the author may need to be clearer when highlighting if something is new or unique in their approach compared to what was already published (apart from the measurement campaign). The model using a regularized vortex cannot really be seen as a new contribution or method. The experimental data though are of high value.*

  It is correct that the method and model is not new. But as we state that the model has its origin in aviation, it is new to be used to describe wind turbine wake vortices properties. We added a paragraph to highlight this new approach in WEC wake studying.

  Also the described method illustrates how to obtain results from in-situ line measurements of a UAS, including systematic measurement uncertainties like skewed/canted vortex hoses etc.

- *The amount of measurements appear unclear, and some statistical analysis and information about the ensemble of results available could be valuable. Reading the article, it seems that only one vortex was analysed. Further data with different distances downstream should be incorporated since according to section 2 more data was acquired. Statistical tools should also be used to mention the uncertainty on the fitted parameters and to quantify the error between the model and the measurements.*

  A new subsection '2.3 Available data' has been added. We also state why only one pair of vortices could be used for an evaluation. A simple vortex measurement in the wake is not sufficient, also the criterion $\Delta y < r_\mathrm{c}$ must be met which is a rare condition. We comment on this in the 'specific comment' section below in more detail.

- *The figures are usually clear. The authors could yet reduce the number of figures, particularly in the first 10 pages, or by combining the measurements with the fitted model in figures 10-17.*

  We will take out some figures from the manuscript. But also some new ones will be added. Mainly to do the uncertainty evaluation (wind angles, true air speed, etc.). We will also try if some figures could be combined. But that might add to the readers confusion.

- *I hope my numerous comments will not discourage the authors, and I strongly encourage them to further work on this paper. As I mentioned earlier, the article has some great potential, it just needs some additional work to reach the point of publication.*

  Highly appreciated emphasis. Thank you!

**Specific comments**

**Abstract**

*The statement in the abstract 'the BH model can be used to describe wake vortices' is probably too strong and would need to be moderated since this simpler model is not capturing all the dynamics. I will comment more on this later.*

We will add the simplifications that have to be made. The BH model is also only a tool to describe the vortex geometry in a 2D cut. This point might need additional stressing.

**Introduction**

*I would think that bringing the context of Germany appears too specific, since the wind energy sector is growing in other countries.*

This is indeed true. The introduction has been 'internationalised'.

*p1 l21: "In research..." this sentence and the following two are hard to read and could be reformulated*

The abstract as well as the introduction were rewritten.

*p2 l8: The scaling problem of wind tunnel measurements could be mentioned here*
Thank you for the hint. It will be mentioned.

*p2 l10: make sure the acronym for UAS (and other acronyms) is made explicit in the introduction*
Has been implemented.

*p2 l15: Could you mention the arguments for the offshore comparison. It probably relies on arguments on the boundary layer when the flow comes from the shore, but wave exciting the turbines and the surface roughness may be different.*
Yes, there is definitely an influence through the surface roughness compared to the sea. It can also be seen in Fig. 5 that the horizontal wind in a certain altitude experiences much more fluctuations. So there is an internal boundary (surface) layer that shows increased turbulence with a more stable marine layer on top.
This was only mentioned since for the briefly presented HeliOW project it would be desirable to fly and measure behind an actual off-shore wind turbine. But for several practical reasons this was simply not feasible.
The influence of turbulence is also a good point for future measurements to get a large amount of vortex measurements to do some statistics.

*p2 l18: "The project aims for safe helicopter flight paths in off-shore wind energy parks" needs reformulation*
Will be changed to "The goal of the project is also to determine safe helicopter flight paths in off-shore wind energy parks".

*p2 l19, l21 \*University of\* Munich, University \*of\* Stuttgart.*
Has been changed accordingly.

*p2 l23: the wind turbine also generates strong coherent vorticies, can these be considered turbulence?*
Coherent (in this case artificial turbulent) structures are not turbulence covered by Kolmogorov 1941. The coherent vortices might be considered as a large scale anisotrop turbulence. The surrounding atmosphere in the best case scenario can be considered as isotrop. In the far wake these coherent structures decay and add to the isotrop turbulence.

*p2 l30: could you highlight more the difference between the study from Subramanian and yours?*
Subramanian et al. detected tip vortices by a pressure signal along a longitudinal flight path. The vortices were also not quantified (vortex strength, core radius). We will add some more details to the manuscript and highlight the main differences.

**Section 2**

*p3 l7: Wind speed and directions could have changed during this 15min period, do you have access to measurements to support this assumption?*
Unfortunately, we didn't have access to highly resolved SCADA data for the measurement time which could have given us precise information about wind speed and direction changes. Moreover, we were flying about 27 m behind a 114 m diameter WEC. If the wind direction would have changed significantly we might have crashed into the rotor blades. For a single vortex measurement it is also not important, if the wind direction varies a bit. The wind velocity component rotation later in the manuscript was done for the single measurement. The mean wind direction is only important to set up the flight trajectory for the UAS.

**Section 3**

*p5 l11: You could mention that the ring vortex is an approximation of the wake vorticity at high tip-speed ratio.*
Thank you for the input. We will mention that.

*p5 l14: This line can be reformulated to mention that this result is true under the vortex ring assumption.*
The line has been modified.

*p6 Figure 4: "recreation" may not be the correct word in the caption, maybe "model", or "reproduction" would be more accurate?*

Done.

*Figure 7: Instead of using north/east for the axis wouldn't it be easier in that case to use an orientation in the frame of the turbine, with y pointing upstream against the main wind direction? You could then remove the sentence at the end of page 6.*

This was actual the intent when planing the flight path at the field campaign. The wind was slowly turning. So Figure 7 shows the actual flight path (direction) of the UAS. So y pointing north is almost into the main wind direction. But at the time of the measurement the wind direction was already a bit off. Also the post-processing software of the UAS data always writes the wind velocities according to the meteorological standards $u$ pointing east, $v$ pointing north and $w$ pointing upwards. So the data in the end always has to be rotated into the main wind direction. And this is necessary to later make $\overline{u'} = 0$ in the wake.

*p7 line1-12: The potential flow assumption probably appears too early in this paragraph and the paragraph could be reformulated. The definition of circulation as function of the vorticity is independent of this assumption. Equation (3) only uses an axi-symmetric assumption. It is yet true that the circulation of a vortex makes more sense in inviscid flows where the vorticity is condensed to confined singular regions.*

We will look into a possible reformulation and implementation of the introduction of potential flow.

*p10 l1-4: This paragraph needs should be reformulated, the language improved.*

The paragraph has been rewritten.

*p10 l9-15 and p11 l1-8: While reading the text I was confused since figure 9b didn't appear to be mentioned. The explanation could be improved by clearly explaining both figures and both scenario, before mentioning figure 10. Alternatively, you could tell the reader to focus only on figure 9a for now and figure 9b will be explained later. Also, equations 8-10 could be introduced first before drawing the conclusions that there is no unique solution. Further, the way the equations are introduced can be improved by telling to the reader what is coming, e.g. "The velocities at point 1 and 2 are...". Right now, they appear in the text announced.*

Figure 9b is mentioned on page 11 l7. I agree that the two different cases of passing a vortex in the UAS measurements needs additional addressing to not lose the reader. The paragraphs have been modified.

*p11 l9: "This double peak can be lead back to passing the maximum tangential velocity at $r = r_c$ at position 1 and 2". The sentence may need reformulation*

Changed to: This double peak is caused by passing the maximum tangential velocity at $r = r_c$ at position 1 and 2".

*p11 l12-15: Similar to the previous remark, the equations can be introduced before drawing the conclusions.*

We think this comes down to personal preference. We would like to explain what's happening when passing the vortex at $\Delta y < r_c$ and follow up with what that means for the previously introduces equations.

*p11: "As shown above the presence and identification of a vortex (or a pair of vortices) is measurable". The language needs to be reformulated (one cannot measure an identification). Also, the previous section seemed to show that in some cases the determination was not possible, which would imply that the identification is not always possible.*

Fair point. "As shown above the presence is measurable and a later identification of a vortex (or a pair of vortices) possible when the $\Delta y < r_c$ is met."

**Section 4**

*p13 l8: It would help the reader to provide some information about the measurement campaigns (how many samples were selected, what were the mean conditions), and some introduction about the samples you selected (and why you selected them) to present in the paper.*

This issue will be addressed with the additional subsection about data availability.

Maybe 'canted' or 'oblique' vortex is the more precise expression. The additional 3D features are meant to help the reader. That is why the blue is more transparent. It could be removed, if it confuses more than it helps?!

We have to introduce the oblique vortex hose, since the real (measurement) world is different from the analytical approximation (ring vortex). We will try to make that more clear.

In Section 3 we want to present the theory/method how to "reconstruct" the vortex from GPS and pressure data. In section 4 the individual fit is calculated. One might consider to just remove this subsection, e.g. Fuertes 2019 never introduces any vortex model of the WEC and just "fitted" the data. As for the second point, in fact the GPS positioning standard deviation is in the range of the vortex core radius. So the double peak is not visible and overall the accuracy is not up to the in-situ, in-line pressure measurement. Thus, to keep an step-by-step approach and to use the analytical model, as a proof of concept, we prefer to keep it separate from the measurement.

You are correct, the statement has been changed. The actual deficit in percent or m/s is only important for the correction of the modelled $v$ component.

**Section 5**

We will try to do so.

The engineering sector of wind energy converters is a highly competitive market. Although the University of Stuttgart has a generic recreation of the wind turbine, it is reserved for wake computations. The resulting aerodynamic properties of the turbine are however strongly confidential. That's why those properties, namely here the thrust coefficient, had to be estimated. As stated on p. 19 l1, we approximated the thrust coefficient vs. wind speed with an NREL 5 MW turbine. Since the E112 and NREL 5MW wind turbines both have similar rotor diameters and produce a similar amount of electricity, the approximation was judged reasonable.

*vorticity surface can be approximated to a vortex cylinder, in which case any trajectory of the drone will indeed cross the tip-vorticity surface twice. This cylindrical surface does not resemble two vortices spinning in opposite direction and the wake dynamics cannot be described by assuming that it consists of two vortices. What the author probably means is that the velocity field across a tip-vortex (or a tip-vorticity surface) resembles the one of a regularized point vortex. This analogy (which is natural given the different analytical vortex wake models of wind turbines) cannot be used to "describe" the wake, but it can be used to "estimate" some of the wake properties, that is, the tip-vortex core radius and circulation.*
We agree. This paragraph needs to be more precise. We will implement your suggestions.

**Section 6**

*p19 l21: The identification of one vortex strength does not appear to be enough to draw a conclusion, or the conclusion needs to be moderated. Also, it may not be necessary to attribute this equation to Sorensen et al. and instead it can be mentioned where this formula comes from: circulation for a rotor of constant thrust coefficient (This should also be mentioned earlier in the text p18 l1-6).*
The conclusions will be readdressed and the statement moderated. Also we want to repeat at this point (and later in the new version of the conclusions) that the fact that we hit two (!) consecutive vortices in one flight leg at the crucial criterion ($\Delta y < r_\mathrm{c}$) might also be considered lucky. That both measurements, after consideration of the non-ideal conditions (canted vortex hose) and therefore the rotation of the horizontal measurements into the vortex plane is giving two circulations strengths that match the theoretical calculation, needs an appropriate (strong) statement.

---

## Author Response (AR2)

[revised manuscript text omitted]
_{\mathrm{t},2} = V_{\mathrm{t},1} = V_{\mathrm{t,max}} = \frac{\Gamma}{2\pi}\,\frac{r_{\mathrm{c}}}{r_{\mathrm{c}}^2 + r_{\mathrm{c}}^2} = \frac{\Gamma}{4\pi r_{\mathrm{c}}} \tag{11}$$

$$V_{\mathrm{t},\Delta\mathrm{y}} = \frac{\Gamma}{2\pi}\,\frac{\Delta y}{r_{\mathrm{c}}^2 + \Delta y^2} \tag{12}$$

$$r_1^2 = r_2^2 = L^2 + \Delta y^2 = r_{\mathrm{c}}^2 \longleftrightarrow \Delta y^2 = r_{\mathrm{c}}^2 - L^2 \tag{13}$$

With now only 3 ($\Gamma$, $r_{\mathrm{c}}$ and $\Delta y$) unknown parameters it is possible to solve the equations.

Dividing Eq. 12 by Eq. 11 eliminates $\Gamma$. Inserting Eq. 13 gives:

$$\frac{V_{\mathrm{t},\Delta\mathrm{y}}}{V_{\mathrm{t,max}}} = \frac{r_{\mathrm{c}}\sqrt{r_{\mathrm{c}}^2 - L^2}}{r_{\mathrm{
[revised manuscript text omitted]

**Author's response to Referee #1**

**July 8, 2019**

Thank you for the detailed review of the manuscript. In the following I will comment on each point. The referee's comments will be repeated in blue italic before the answer.

**General comments**

*The manuscript by Mauz et al. describes in-situ measurements of wind turbine tip vortices with a UAS. From these measurements, circulation of the vortices is calculated using the Burnham-Hallock wake vortex model. These measurements are unique and I do not know of any other study in which a UAS operated in such close proximity to an operating wind turbine and even in its wake. The authors can convincingly show that wake vortices can be measured with the UAS. There are still some issues in the revised manuscript that I think need to be taken care of.*

**Specific comments**

**Abstract**

*p.1,ll.11f: "Also proposition..." Something is wrong with the grammar here.*
Thank you, the grammar issue has been addressed.

*p.1,l.13: What is "overall turbulence"?*
The meaning of "overall" in this case is "the cumulative, surrounding, atmospheric" turbulence. We changed it to "overall, atmospheric turbulence" to distinguish it from the sole turbulence generated by the WEC.

**Introduction**

*p.1,l.18: recommendation: "... the capacity of a single turbine"*
The sentence has been changed accordingly.

*p.2,l.2: "Not least, because of improvements ..." - please revise.*
Has been changed to:
Once measured data has revealed some possible tweaks and enhancements to a model, improvements can be made and flow back into the (e.g. numerical) model.

*p.2,l.12: There are also short-range continuous-wave lidars which have significantly higher spatial resolution for short focal distances and have been applied for WEC wake measurements. These measurements can still not resolve the tip vortices, but should be mentioned for completeness. See e.g.:*
*Menke, R., Vasiljevic, N., Hansen, K. S., Hahmann, A. N., and Mann, J.: Does the wind turbine wake follow the topography? A multi-lidar study in complex terrain, Wind Energ. Sci., 3, 681-691, 2018. Herges, T. G., Maniaci, D. C., Naughton, B. T., Mikkelsen, T. K., & Sjöholm, M. (2017). High resolution wind turbine wake measurements with a scanning lidar: Paper. In Wake Conference 2017 (Vol. 854). Journal of Physics: Conference Series*
Thank you for the valuable input. The paper, mentioned above, has been added to the reference list.

[Figure]

Figure 1: Weather situation of the day of the measurement July 24 2018 at 6 PM.

**Section 2**

*p.3,l.5: I think the link to Pixhawk should be given as a proper reference.*
The reference has been implemented.

*p.5,Fig.3: Please label or mark the WEC more clearly in the picture.*
An updated graphic, containing a clear indication of the WEC, is now being used.

*p.5,l.6: You state that only a single flight leg is presented, but then say the presented data was captured within 15 minutes. I do not think the one flight leg took that long, so pleases be more precise in the description of the data. The TKE seems very low, even for a quasi-marine boundary layer. Can you please add information about the time of day and any kind of information about the synoptic or mesoscale situation. You say that TKE was calculated from a ten second measurement which would be way too short for a point measurement. I assume you calculated it along a flight leg, but you have to let the reader know, including the length of the leg.*

We have now implemented more information about the flight leg and set it apart from the flight pattern that took 15 min.
The flight took place approximately at 6.30 PM at a usual summer day with clear sky and $25°C$ air temperature. The TKE calculation has been done at hub height in the undisturbed, free atmospheric flow, averaged over approximately $200\ m$ distance. Regarding the relatively low value for TKE: Depending on the calculation window ($\pm100$ indices) the TKE is about 0.1 m²s⁻². This corresponds to a standard deviation of the horizontal wind of $\approx 0.3$ m²s⁻² which should be alright for an evening transition with decreasing turbulence.

To clarify the mesoscale weather situation we can provide weather maps of the day of the field campaign (Fig. 1) and if needed, implement it to the manuscript.

**Section 3**

The sentence has been altered.

**Section 4**

Thank you, their position has been corrected.

*p.14,l.14f: Changing the name of the new coordinate system is not enough. There is still an explanation of the new coordinate system required and this should probably go to section 3, because it is a method and not a result. In line 22 it is still called vortex coordinate system by the way.*

We want to argue that this data rotation into the vortex plane is less a method than a reaction to the measurement reality, since the method assumes a ring vortex instead of a helical vortex pattern. It ought to be understood as the UAS canting into the vortex rotational plane to catch the whole rotational energy in a 2D-plane, whatever its orientation, and therefore minimising the $w$ component. It is here described as the transformation of the UAS coordination system into the vortex coordination system (e.g. in line 22). This (individual) transformation introduces an error in the residual time series, but at a distance of $\pm r_c$ around the vortex core where the evaluation takes place, the data is corrected. Meaning, the purple dashed and purple solid line overlay.

The section has been re-written and is now better structured. The purpose of this data rotation has been explained in separate paragraphs, followed by the presentation and explanation of the figures/data.

A more detailed description has been added.

Thank you, I can understand how this might be confusing. So we have to be more clear hear. Therefore, I want to distinguish between to kinds of "out of range". The more severe type of "out of range" measurement is, when the 5-hole probe gets is measuring extreme pressure fluctuations due to the tip vortex. Sometimes those angles (and pressures) are so steep and and the pressures very high, that the pressure sensor itself is simply out of range. As a consequence, the data shows NAs or a data lag in the time series, when plotted. These data can not be used, since a part of the single most important feature, the vortex measurement, is missing.
Then, there exist measurements, where the 5-hole probe is exposed "near critical". The flow angles are steep, but the pressure sensor can still handle it. No data lags are produced. Usually the flow angles are interpolated inside the calibrated range. But when a measurement is outside of the calibrated range, the flow angle can be extrapolated using the ninth order polynomial equation.

**Section 5**

We can see the point. The sentence has been rephrased.

**Section 6**

As shown in the section covering the measurement error, the measurement uncertainty is about 5 to 10 %. We will be more precise on p.20,l.21. And still, only for high variations of the true air speed.

This is correct, different 5-hole probe geometries excel at different "measurement environments". The 5-hole probe operating on the MASC-3 UAS allows for an extension of the calibration range to $\pm 40°$.

The position of the UAS is used for the analytical BH model, e.g. Fig. 17. For future evaluation the position of the vortex can be calculated. Possible vortex or wake meandering could be derived from these positions, also the widening of the core radius can then be well documented.

---

## Author Response (AR3)

**Author's response to Editors comments**

July 24, 2019

Dear Sandrine Aubrun,
thank you for the detailed review and editing of the manuscript. Below I will comment on each point.
Your comments will be repeated in blue italic before the answer.
I hope all issues have been addressed to your satisfaction.

Sincerely,
Moritz Mauz

**General comments**

*Could you please add a few sentences on the "two kinds of out-of-range" issues on the paragraph 2.1, to remove the confusion noted by reviewer 1.*
Thank you, we also see the need for a clarification. You suggest to clarify the "out of range" topic in section 2.1 (Research Aircraft). Maybe this is a mix-up of numbers, but, we suggest section 2.3 (Available data), since this issue directly affects the usability of the data. Section 2.3 has been updated with an additional paragraph to cover this topic.

*Add a reference to justify that "the 5-hole probe operating on the MASC-3 UAS allows for an extension of the calibration range to $\pm 40°$".*
A reference has been added in which a similar (conical) 5-hole probe has been calibrated for flow angles up to $\pm 40°$. In the conclusion of the manuscript it has also been clarified what the results of these extended angles would be.

*Remove the comma between "overall" and "atmospheric" in page 1, line 14.*
The comma has been removed.

[revised manuscript text omitted]